

# Thermal and near-infrared sensor for carbon observation Fourier-transform spectrometer-2 (TANSO-FTS-2) on the Greenhouse Gases Observing Satellite-2 (GOSAT-2) during its first year on orbit

Hiroshi Suto[1], Fumie Kataoka[2], Nobuhiro Kikuchi[1], Robert O.Knuteson[3], Andre Butz[4], Markus Haun[4], Henry Buijs[5], Kei Shiomi[1], Hiroko Imai[1] and Akihiko Kuze[1]

[1]Japan Aerospace Exploration Agency, Tsukuba-city, Ibaraki, 305-8505, Japan
[2]Remote Sensing Technology Center of Japan, Tsukuba-city, Ibaraki, 305-8505, Japan
[3]University of Wisconsin-Madison, Madison, WI, 53706, USA
[4]Institut für Umweltphysik, Universität Heidelberg, 69120 Heidelberg, Germany
[5]FTS Consulting, Quebec, G3E 1H7, Canada

*Correspondence to*: Hiroshi Suto (suto.hiroshi@jaxa.jp)

**Abstract.** The Japanese Greenhouse gases Observing SATellite-2 (GOSAT-2), in orbit since 29 October 2018, follows up
the GOSAT mission, itself in orbit since 23 January 2009. GOSAT-2 monitors carbon dioxide and methane in order to increase our understanding of the global carbon cycle. It simultaneously measures carbon monoxide emitted from fossil fuel combustion and biomass burning, and permits identification of the amount of combustion-related carbon. To do this, the satellite utilizes the Thermal and Near Infrared Sensor for Carbon Observation Fourier-Transform Spectomer-2 (TANSO-FTS-2). This spectrometer detects gas absorption spectra of solar radiation reflected from the Earth's surface in the
shortwave-infrared (SWIR) region as well as the emitted thermal infrared radiation (TIR) from the ground and the atmosphere. TANSO-FTS-2 can measure the oxygen A band (0.76 um), weak and strong $CO_2$ bands (1.6 um and 2.0 um), weak and strong $CH_4$ bands (1.6 um and 2.3 um), a weak CO band (2.3 um), a mid-wave TIR band (5.5-8.4 um) and a long-wave TIR band (8.4-14.3 um) with 0.2 cm$^{-1}$ spectral resolution. TANSO-FTS-2 is equipped with a solar diffuser target, a monochromatic light source, and a black body for spectral radiance calibration. These calibration sources permit
characterization of time-dependent instrument changes in orbit. The onboard-recalibrated instrumental parameters are considered in operational level-1 processing and released as TANSO-FTS-2 level-1 version 102102 products, which were officially released on 25 May 2020. This paper provides an overview of the TANSO-FTS-2 instrument, the level-1 processing, and the first-year on-orbit performance. To validate the spectral radiance calibration during the first year of operation, the spectral radiance of the version 102102 product is compared at temporally coincident and spatially collocated
points from February 2019 to March 2020 with TANSO-FTS on GOSAT for SWIR, and with AIRS on AQUA and IASI on METOP-B for TIR. The spectral radiances measured by TANSO-FTS and TANSO-FTS-2 agree within 2 % of the averaged bias and 0.5 % standard deviation for SWIR bands. The agreement of brightness temperature between TANSO-FTS-2 and



AIRS, IASI, is better than 0.5 % in the range from 220 K to 320 K. GOSAT-2 not only provides seamless global $CO_2$ and

$CH_4$ observation but also observes local emissions and uptake with an additional CO channel, fully customized sampling

patterns, higher signal-to-noise ratios, and wider pointing angles than GOSAT.

## 1 Introduction

Measurements of the spatial distribution, temporal variations, and trends of carbon dioxide ($CO_2$) and methane ($CH_4$) in the

atmosphere have played and will continue to play a key role in the elucidation of their atmospheric budgets (UNFCC, 2015).

For the last decade, space-based measurements have become a powerful tool for generating global distribution maps of $CO_2$

and $CH_4$. The Japanese Greenhouse gases Observing SATellite (GOSAT) (Kuze et al., 2009), named IBUKI, in orbit since

23 January 2009, the Orbiting Carbon Observatory-2 (OCO-2) in orbit since 2 July 2014 (Crisp et al., 2004, 2008, 2017), and

the Sentinel-5 Precursor/TROPOMI (S5P) in orbit since 13 October 2017 (Hu et al. 2018) monitor the column-averaged $CO_2$

dry-air mole fraction, $X_{CO2}$ (GOSAT and OCO-2), and the column-averaged $CH_4$ dry-air mole fraction $X_{CH4}$ (GOSAT and

S5P). These observations have been ongoing along with public distribution of the data for over 11 years, 5 years, and 2

years, respectively.

The design life of GOSAT was five years but it has remained operational for over eleven years. Long-term monitoring

by a single well-calibrated and characterized instrument has the advantage of quantifying year-by-year global and regional

concentration changes. To maintain long-term monitoring with uniform observation quality, it is essential to establish the

same standard of measurement performance for follow-up missions (Kataoka et al., 2017). To provide continuous

monitoring of the global distribution of $X_{CO2}$ and $X_{CH4}$ and to continue the success of the GOSAT mission, GOSAT-2 was

launched on 29 October 2018 and carried the Thermal And Near infrared Sensor for carbon Observation Fourier-Transform

Spectrometer-2 (TANSO-FTS-2) and the Cloud and Aerosol Imager-2 (TANSO-CAI-2). Several functionalities of the

GOSAT-2 instruments are improved over those of the TANSO-FTS and TANSO-CAI employed on GOSAT.

GOSAT-2 can measure the ShortWave InfraRed (SWIR) solar radiation reflected from the earth's surface as well as the

Thermal InfraRed (TIR) radiation from the ground and the atmosphere. The TANSO-FTS on GOSAT can cover a wide

spectral range, specifically, three narrow bands (0.76, 1.6, and 2 um) and a wide band (5.5-14.3 um) with 0.2 $cm^{-1}$ spectral

resolution. In order to surpass the GOSAT observation capability, GOSAT-2 has extended SWIR spectral coverage; one

extension is toward the shortwave for solar-induced fluorescence, and the other is toward the longwave for carbon monoxide

(CO) in the 2.3 um region (also covered by S5P). Also, TIR spectral coverage is divided into two regions, band 4 (5.5-8.6

um) and band 5 (8.6-14.3 um). TANSO-CAI-2 on GOSAT-2 is an imaging radiometer for the ultraviolet (UV), visible, and

SWIR regions providing cloud and aerosol information.

Characterizing the instrument in orbit is essential to providing spectral radiance products that are seamless from

GOSAT to GOSAT-2 as well as providing consistency with other satellites, such as OCO-2 and S5P. During GOSAT-2's

first year of operation, several calibration processes for characterizing TANSO-FTS-2 were carried out with onboard



calibrators. Also, GOSAT-2 spectral radiance products are compared with other satellite data such as TANSO-FTS on
GOSAT, AIRS on AQUA (Aumann et al., 2003) and IASI on METOP-B (Clebaux et al., 2009). Finally, we will show that
the spectral radiance for GOSAT-2 is consistent with these satellites' inter-calibration data.

This paper first introduces an overview of the GOSAT-2 satellite and the TANSO-FTS-2 and CAI-2 instruments. The
processing method for transforming raw on-orbit data to calibrated spectral radiances (level-1 processing) for TANSO-FTS-
2 follow. Next is an assessment of the first year of in-orbit performance of TANSO-FTS-2 by its comparison to temporally
and spatially coincident data from other satellites. In addition, calibration challenges are identified with current best
estimated values.

## 2 GOSAT-2 overview

### 2.1 GOSAT-2 satellite system

GOSAT-2 monitors $CO_2$, $CH_4$, and CO globally from space. GOSAT-2 is placed in a 613 km sun-synchronous orbit at 13:00
local time, with an inclination angle of 98º and a revisit time of six days. Figure 1 shows a prelaunch image of GOSAT-2.
The orbital parameters for GOSAT-2 are listed in Table 1 as well as those for GOSAT.

GOSAT-2 carries two instruments: the TANSO-FTS-2 and the TANSO-CAI-2. Section 3 focuses on the TANSO-FTS-
2 spectrometer in detail.

To identify clouds and aerosol, GOSAT-2 carries a forward- and backward-viewing imager, the TANSO-CAI-2. It
covers a broad (1000 km) cross-track field of view to facilitate measurements of aerosol and cloud conditions. TANSO-CAI-
2 observes up to +20 deg forward in the 343 nm, 443 nm, 674 nm, 869 nm, and 1630 nm bands and up to -20 deg backward
in the 380 nm, 550 nm, 674 nm, 869 nm, and 1630 nm bands.


### 2.2 Operation

Normal scanning, including fixed grid and target observations, and pointing at sun-glint over the ocean, is carried out
according to a set pattern and is determined prior to the observation cycle. TANSO-FTS-2 accommodates a fully
programmable pointing system to extend observation control capabilities. This system allows operators to point to any
location on the earth's surface to observe, and to manage the observation locations day by day. Target observations for the
monitoring megacity emissions and validation sites are planned ahead of time. A particular orbital trajectory that facilitates
an exact overflight of Lamont, OK (Latitude 36.60 North, and Longitude 97.49 West) is selected for the purpose of routine
validation. An observation interval of 4.024 seconds and a nominal turnaround time of 0.65 seconds are employed for each
observation. The daily plan of observation locations is managed in the Mission operations Management Organization
(MMO). Due to safety concerns with the pointing mirror scanner motor, the difference of the optical angles between two



consecutive observations must be less than 25.5⁰. If the motor receives an angle command greater than 25.5⁰, TANSO-FTS-2 immediately transits from normal operation to safe mode, suspending observations. Nominally, the observation plan is uploaded to the satellite once a day. The observed data is recorded by the onboard data recorder and is transferred periodically from the satellite to ground typically once every two orbits.


## 3 The TANSO-FTS-2 instrument

### 3.1 Instrument overview

The greenhouse gases sensor takes spectroscopic measurements of shortwave infrared sunlight reflected from the earth's surface to the satellite and TIR radiation emitted from the ground and the earth's atmosphere. The gaseous column amounts

(column abundances) of $CO_2$ and $CH_4$ in the 1.6 um band, $CO_2$ in the 2.0 um band, and $CH_4$ and CO in the 2.3 um band are estimated, and the gaseous concentration profiles (vertical distribution) are assessed using the TIR region. A Fourier transform spectrometer is used because it allows simultaneous observations with a high spectral resolution ($0.2 \text{ cm}^{-1}$) over a wide bandwidth. An interferogram measurement created in 4.024 seconds is taken as a baseline. TANSO-FTS-2 uses almost the same type of Fourier-Transform Spectrometer (FTS) mechanism as the TANSO-FTS, and its main characteristics are

summarized in Table 2. The optical layout is shown in Fig. 2.

A two-axis pointing control mechanism allows the sensor to aim at planned locations for a complete interferogram measurement interval after which it moves to the next observation location. It can also observe sun glint over oceans. The two-axis pointing system for TANSO-FTS-2 has wider coverage than that for TANSO-FTS and provides fully programmable aiming to up to 1246 observation points per orbit. It allows between -40⁰ and +40⁰ in the along-track direction

and between -35⁰ and +35⁰ in the cross-track direction. In addition, to keep the lubricant in each bearing uniform, wide travel motion in both along-track and cross-track directions are scheduled once per day.

The input optics, interferometer, band separation optics, and the detector optics are housed in a single, temperature-controlled  optical box. The scene flux is reflected into the input optics by a bare gold-coated mirror on the two-axis pointing system, and it is divided into two parts by a pickoff mirror. One beam is directed to the CMOS video camera (608 x 1024

pixels) for identifying the scene image, the other to the interferometer (FTS). The camera image is also used to identify cloud positions in the field of view. To reduce the number of cloud-contaminated observations, the TANSO-FTS-2 uses the cloud images to actively avoid clouds by adjusting the line of sight during the FTS turnaround motion just before the measurement scan.

TANSO-FTS-2 employs a double pendulum and double cube corner type of FTS mechanism with an uncoated ZnSe

beam splitter (as for TANSO-FTS). The effective aperture size is 73 mm, which is larger than that of TANSO-FTS, and the maximum optical path difference is +/- 2.5 cm. A long-life diode laser acts as a metrology light source, with a stable single-





mode 1.31 um emission, in order to last through the five-year mission. The interferometer temperature is controlled to 23 +/-3 deg C. The temperature of the blackbody ranges between 293 K and 295 K during the first year's operation.

The scene flux signal is sampled by the FTS at fixed time intervals and measures both the science signal and laser fringe signal simultaneously; this is called uniform time sampling. Many conventional FTS mechanisms, including TANSO-FTS, use a laser fringe triggered measurement technique called uniform optical-path-difference sampling. In level-1 processing, the uniform time-based interferogram is converted to a uniform optical-path-difference sampled interferogram, followed by the inverse FFT (Fast Fourier Transform). The detailed processing method is described in the following section.

The modulated scene flux is divided into the various SWIR and TIR bands by dichroic beam splitters. The separated beams are focused on three kinds of detectors, a silicon(Si) detector for the $O_2$ band 1, PhotoVoltaic - Mercury Cadmium Telluride (PV-MCT) detectors for SWIR bands 2 and 3, and for the shortwave TIR band 4, and a Photo Conductive - Mercury Cadmium Telluride (PC-MCT) detector for the longwave TIR band 5. A multistage passive space cooler provides detector cooling. The detector temperatures are set at 215 K for band 1, 130 K for bands 2 and 3 and 100 K for bands 4 and 5, respectively. The field of view for all the bands is limited to 15.8 mrad. The instantaneous ground field of view (GIFOV)

becomes a 9.6 km in diameter circle at the sub-satellite location from an altitude of 613 km.

Simultaneous observations of two linear polarizations for the 0.76 um, 1.6 um, and 2.0-2.3 um bands are facilitated by two identical detectors coupled with a polarizing beam splitter. TANSO-FTS-2 has a somewhat extended spectral coverage compared to TANSO-FTS. The TANSO-FTS-2 spectral regions, which are defined by bandpass filters, are as follows: band 1 (12950–13250 cm$^{-1}$), band 2 (5900–6400 cm$^{-1}$), band 3 (4200–5200 cm$^{-1}$), band 4 (1188–1800 cm$^{-1}$), and band 5 (700–

1188 cm$^{-1}$). The signal voltages from the detectors are converted into numerical representations by 14-bit analog-to-digital convertors (ADCs). Just before an ADC, the DC component of the offset science signal is removed, but its value is included in the data so that the ADC handles only the full dynamic range of the AC component. Also, high frequency pulses are counted for each laser fringe interval as reference times for the optical path difference.

**3.2 Calibration operation**

Solar irradiance calibration (Sol. Cal.), deep space calibration (DS Cal.), and blackbody calibration (BB Cal.) are all collected over each orbit using a set timing pattern. Nadir observations are not made during the DS Cal. or BB Cal. period. If a DS Cal. and BB Cal. are to be managed at a fixed latitude, observations of certain latitudes would lead to a loss of nadir observations at certain locations. For this reason, the DS Cal. and BB Cal. are dispersed over the orbit to facilitate a uniform

data acquisition coverage that is the same as for TANSO-FTS. TANSO-FTS is also operated six calibrations per orbit – twice during the day side and four times at the night side.

The Sol. Cal. uses a solar diffuser activated with a shutter: 17 seconds before the start of calibration measurements, the shutter opens and the primary solar diffuser is exposed to sunlight. In addition, a secondary solar diffuser is used for



reference operations, which permits monitoring of the degradation of the reflectance of the primary diffuser. Reference
diffuser operations are scheduled once every three months.

A lunar calibration that complements the solar diffuser calibration is also done, once a month. The Instrumental Line
Shape function (ILS) calibration is performed using two types of diode laser; 0.77 um and 1.54 um diffused via an
integrating sphere to illuminate the full field of view of the interferometer. The schematic diagram of a typical calibration
operation for one orbit is shown in Fig. 3.


### 3.3 Instrument models

The initial characterization for TANSO-FTS-2 was done during prelaunch testing and the calibration phase. Generally,
retrievals of the column abundances of $CO_2$, $CH_4$, and CO require instrument models that are compatible with the retrieval
software. Auxiliary information such as radiance conversion factors, signal-to-noise-ratio (SNR) models, and ILS models is
available via the GOSAT-2 product archive site (https://prdct.gosat-2.nies.go.jp/en/document.html).

### 3.3.1 Radiance conversion model

For prelaunch radiance calibration, the output signal level of TANSO-FTS-2 was compared to the radiance levels of a 65-
inch integrating sphere whose inner surface was coated with barium sulfate. The radiance levels emitted by the integrating
sphere were characterized based on an NIST standard lamp. The calibration was conducted in a temperature- and humidity-
controlled room, but the instrument was not installed in a vacuum chamber during the radiance calibration period.

TANSO-FTS-2 is equipped with a passive space cooler and with heaters to control the temperatures of all detectors as
well as the optical components. Under laboratory conditions, the passive cooler could not provide power enough sufficient to
cool the detectors to their set point temperatures. To assist in cooling, an additional external cooler system was coupled with
the TANSO-FTS-2 passive cooler during prelaunch calibration.

The instrument was illuminated by the integrating sphere with both along-track and cross-track pointing angles at zero
deg. Due to the presence of oxygen and water vapor lines in bands 1 and 3, the envelope of the spectrum is assumed to
determine the radiance conversion coefficients. The conversion coefficients from raw spectra to radiance are available on the
GOSAT-2 data distribution site (https://prdct.gosat-2.nies.go.jp/en/document.html). In level-1 products, the radiance spectra
are processed by applying these conversion coefficients. However, the two-axis orientation dependency of the scanner mirror
reflection is not taken into account in the current level-1 products. In orbit, the radiometric response for the SWIR bands has
been changed and were recalibrated based on an on-orbit calibration dataset. The details are discussed in section 5.2. In
addition, the conversion coefficients are revised by the first-year coefficients.





### 3.3.2 Polarization sensitivity model


TANSO-FTS-2 provides both p and s linear polarization data, just as TANSO-FTS does. The measured radiances $I_P$ and $I_S$ are related linearly to the Stokes vector components $I$, $Q$, $U$, and $V$ by a 2 x 4 matrix according to equation (1). (O'Brien et al., 2013.)

$$\begin{pmatrix} I_P \\ I_S \end{pmatrix} = \begin{pmatrix} mPI & mPQ & mPU & mPV \\ mSI & mSQ & mSU & mSV \end{pmatrix} \begin{pmatrix} I \\ Q \\ U \\ V \end{pmatrix} \tag{1}$$

The elements of the 2 x 4 matrix are identified as the polarization model, the Mueller matrix, for TANSO-FTS-2. To characterize the polarization model for TANSO-FTS-2, the instrumental response for each polarization band was characterized by using linearly polarized light during the prelaunch test and calibration periods.

A linear polarizer between the instrument aperture and the integrating sphere was rotated in 10 deg steps while the TANSO-FTS-2 acquired interferograms. These interferograms were analyzed and processed with a polarization model. Generally, the V component in equation (1) is negligible for atmospheric composition measurement. Then, a 2 x 3 matrix dependent on wavenumber is processed against the $I$, $Q$, and $U$ components.

Figure 4 presents the estimated polarization characteristics based on the prelaunch measurements. To implement the

matrix, the retrieval teams have to validate these sensitivities by evaluating their forward calculations (Butz et al., 2011, Kikuchi et al., 2016, O'Dell et al., 2012, Yoshida et al., 2011, Parker et al., 2011, Heymann et al., 2015, Oshcheplov et al., 2013) for particularly polarization-sensitive scenes such as sun-glint data (O'Brien et al., 2013). The implementation and fine-tuning of the 2 x 3 matrix needs to be carried out by a separate level-2 algorithm. The prelaunch Mueller matrix is available from the GOSAT-2 data distribution site as a reference.


### 3.3.3 Signal-to-noise ratio (SNR) characterization

The signal-to-noise ratio (SNR) for TANSO-FTS-2 is determined by the ratio of in-band signal intensity divided by out-of-band signal intensity. The maximum resolving power achievable with the field of view of the FTS configuration is 32000:1, and the achievable resolution in band 1 is < 0.4 cm$^{-1}$. As supplemental information for level-2 processing, the SNR for each

observation is stored in the operational level-1 products. Figure 5 presents a typical SNR for TANSO-FTS-2.

The temperature of the interferometer for TANSO-FTS-2 is controlled to around 295 K. As a result, the thermal background radiation is not negligible and needs to be calibrated with BB and DS calibration. However, the actual signal, which represents the radiation from the earth and the thermal emission of the instrument itself, has to be balanced. In the TANSO-FTS-2 case, the balanced temperature that creates an almost zero-amplitude interferogram condition coupled with





the earth radiation and instrument internal thermal emission, for TIR is around 220K. For this reason, the calculated SNR for TIR with this procedure presents a quick reference of incoming signal intensity.

All bands have 16 signal gain steps. The gain is changed over high-reflectance areas. The nominal gains are assigned in steps of 13, 12, 8, 8, 12, 13 for bands 1p, 1s, 2p, 2s, 3p, and 3s, respectively. The gains for bands 4 and 5 are fixed as steps 7 and 8, respectively.

The simplified SNR is included in the operational L1B products as auxiliary information for level-1 users. It is based on the ratio between the maximum (max) of the in-band signal and the standard deviation (std) of the out-of-band signal (at the lower and the upper ends of the spectral range). The formulation of the simplified SNR is expressed by equation (2).

$$Simplified\ SNR = \frac{\max(inband\ signal)}{average(std(lower)+std(upper))},$$  (2)


In addition to the simplified SNR estimate, the full SNR model is given by equation (3).

$$SNRmodel = \frac{x-c}{\sqrt{a^2+b^2(x-c)}},$$  (3)

where $x$ is the monochromatic radiance and the empirical parameters $a$, $b$, and $c$ are listed in Table 3.

### 3.3.4 ILS model

The ILS function is also characterized during prelaunch testing and the calibration period. The monochromatic light sources for bands 1 and 2 are used for this test. In addition, gas cells filled with $CO_2$, and CO are used for bands 3, 4, and 5. The

modelled ILS functions are processed according to these measurements and issued as the initial versions of ILS functions on the GOSAT-2 web site.

The in-orbit ILS functions are expected to be different than the test set because of small optical alignment changes that may have happened during and after launch. Updated ILS functions were determined through comparison between on-orbit ILS calibration and solar calibration datasets. The most recent estimated ILS function models are plotted in Fig. 6. A later

part of this paper will discuss the challenges related to acquiring ILS knowledge.

### 4 Level-1 processing algorithm for TANSO-FTS-2

TANSO-FTS-2 acquires the uniform-time-sampled interferogram for each observation and each detector channel using a high rate of sampling to minimize noise. The uniform-time-sampled interferogram is numerically filtered and decimated,





which have a factor 5 for bands 1 and 2, a factor 6 for band 3, a factor 12 for bands 4 and 5, in real time to reduce the data volume needing to be stored in the onboard mission data record processer (MDP), and transmitted to the ground once every two orbits.

The first step of processing in the ground facility is generating the level-0 product. Level-0 processing consists of data sorting and decompressing raw data. The second step is to generate level-1A/UTS files, which means that Uniform Time-
sampled (UTS) interferograms (1A) are created combining a uniform-time-sampled interferogram and ancillary data such as satellite position, line of sight, and housekeeping data for the instrument.

Atmospheric spectral radiances are retrieved from the uniform-time-sampled interferograms by applying the level-1 processing algorithm. The level-1 processing algorithm is composed of three main modules: (1) transform from uniform-time-sampled interferogram to uniform optical path difference (OPD) interferogram; (2) apply Fourier transform from SWIR
interferograms to SWIR spectra (bands 1, 2 and 3); and (3) apply Fourier transform from TIR interferograms to TIR spectra (bands 4 and 5). Figure 7 shows the processing overview of both SWIR (Fig. 7(a)) and TIR (Fig. 7(b)).

## 4.1 Processing for uniform-time-sampled to uniform OPD-sampled interferogram.

All the processing from interferogram to atmospheric radiance spectra is performed on the ground. In the first step, the
decimated uniform-time-sampled interferogram is constructed with a DC offset and gain correction. The equation for the uniform-time-sampled interferogram is described by equation (4).

$$I^X_{amp,b,d} = \frac{ADC\_scale_b}{PGA_{gain}{}^X_b} \cdot DN^X_{b,d} \cdot + DAC_{scale\,b} \cdot DC_{offset}{}^X_{b,d} + V_{offset,b} \,, \tag{4}$$

where,

| | |
|---|---|
| $b$: | Bands (bands 1P, 1S, 2P, 2S, 3P, 3S, 4, 5) |
| $d$: | Scan direction (Forward=1, Backward=0) |
| $X$: | Observation target (nadir obs., Black body Cal., Deep Space Cal., Solar Cal., Lunar Cal., ILS Cal., Dark Cal.) |
| $I^X_{amp,b,d}$: | Uniform-time-sampled interferogram with DC offset and gain correction applied. |
| $ADC\_scale_b$: | ADC conversion scale |
| $DN^X_{b,d}$: | Digital number for each interferogram |
| $PGA_{gain}{}^X_b$: | Gain factor for each band |
| $DAC\_scale_b$: | Digital-to-analog conversion factor for each band |
| $DC\_offset^X_{b,d}$: | DC offset clamped at observation start period |







$V_{offset,b}$:          Offset signal.

If the interferogram value at zero path difference (ZPD) is equal to or greater than the full bit range for a band, then a quality warning flag is set for that band, since interferograms with data saturation are not suitable for data processing.

However, since the saturation is identifiable by setting a flag, the data will be inverse Fourier transformed and stored as part of the L1B data.

When passing through the South Atlantic Anomaly, spikes may occur in the interferograms due to energetic particle radiation. If a spike is included in the interferogram, then a quality (warning) flag is set in the L1B file. Theoretically, a one-point spike signal creates a single, generally high-frequency noise component over the complete spectral domain after the

inverse Fourier transform. To reduce this noise, a spike signal correction technique is applied. In the case of a spike being at the $N$ -th point, the $N$ -th sampled data is replaced by the average value of samples $N-1$ and $N+1$. In the case of the edge of the interferogram, $N = 2 \; or \; N = end - 1$ sample is used.

The acquisition duration for time-sampled interferograms, called the sampling window, is synchronized with the scan motion of the interferometer. When the metrology pulse is active, the counter of the sampling clock is incremented by one

for each metrology pulse, then the counted values are transmitted to the MDP as the time interval between metrology pulses. The nominal count is 3458 sampling clock pulses. In parallel, the nominal sampling frequency of the SWIR and TIR signal channels is 117 kHz. The observed interferometric signal is converted from analog to digital by the ADC, which is triggered by the master clock. The original ADC samples are called undecimated samples. The delay between metrology pulses and the science signal is not zero, therefore the time delays have to be included by the processing algorithm. In the nominal

observation mode, the signal is decimated with specific values. The decimated science signal is transmitted to the ground and retrieved for atmospheric spectra. In parallel, the metrology time data is also transmitted as counts to the ground. The number of sampling pulses of the metrology signal is fixed at 76789: Two sample pulses are generated for each OPD change of 1 wavelength of the metrology laser so, at a wavelength of 1.31um, the number of sampling pulses correspond to a total OPD change of 5 cm.

To process OPD-sampled science interferograms, time-sampled science interferograms are coupled with the time-sampled metrology signals, because the metrology signal contains both time and OPD domain information. OPD-sampled interferograms are obtained by applying a digital filter to the time-sampled interferogram in ground processing. The coefficients of digital filters are optimized during prelaunch tests.

Bands 2, 3, and 4 use PV-MCT detectors, and band 5 uses a PC-MCT detector. MCT detectors generally have a

nonlinear response that is normally more pronounced with PC-MCT detectors. If the nonlinearity is not negligible, a wider range of interferogram frequencies should be considered. Theoretically, the parent signal due to nonlinearity shows the harmonic signal features in out-of-band regions. Then, the nonlinear correction terms are characterized with verifying out-of-band signal intensity. For bands 2, 3, and 4, there are no observed harmonic signal features in the out-of-band regions. Only band 5 for the TANSO-FTS-2 has been implemented with a second-order nonlinear term in operational processing.




## 4.2 Processing for bands 1, 2, and 3 spectra

This section provides an overview of how OPD-sampled interferograms (referred to as interferograms here) are transformed into spectra. The basic processing steps are the same as described for the TANSO-FTS per Kuze et al., 2012, 2016. Figure 7 (a) is a schematic diagram for the processing flow for the SWIR bands. Generally, the optical and mechanical ZPD positions

are not the same. To identify the optical ZPD position, the samples of maximum signal around the sampling center are detected first. Then the fringe count error (FCE) is calculated and applied to the correction of the ZPD position. The FCE is calculated using an interferogram truncated around the maximum signal position with a given truncation size. To correct for the phase error, the Mertz method is employed along the following steps. First, the complex spectrum without phase correction $S_{full,b,d}$ is calculated by


$$S_{full,b,d} = FFT\big[fftshift(I_{b,d})\big], \tag{5}$$

where

$FFT$:            Operator for Fourier transform

$fftshift$:    Operator for shifting the zero-frequency component to the center of the array.

Then the phas-corrected spectrum $S_{b,d}$ is given by equation (6)

$$S_{b,d} = S_{full,b,d} \cdot exp\big[-i \cdot angle(S_{Low,b,d})\big] \cdot dOPD_b , \tag{6}$$


and the associated wavenumber $\sigma$ is given by equation (7)

$$\delta\sigma = \frac{1}{dOPD_b} , \tag{7}$$

Where

$S_{Low,b,d}$:   Low resolution spectrum filtered with a gauss function for phase correction

$dOPD_b$:   Sampling interval in inverse wavenumber units





Next the phase-corrected spectra, $S_{b,d}$, in units of V/cm$^{-1}$ are converted to radiance units through multiplication by the

radiance conversion coefficients, $CNV_b$ as well as the time-dependent degradation factor: $Y(v, t)$. The radiance spectra at

time $t$ and wavenumber $v$ are then given by equation (8).

$$L_{b,d}(v, t) = \frac{CNV_b \cdot S_{b,d}(v,t)}{Y(v,t)},$$    (8)

**4.3 Processing for bands 4 and 5 spectra**

In contrast to the SWIR (band 1-3 spectra) processing, TIR processing requires not only nadir (earth scene) observation

spectra but also calibration spectra both of deep space and black body calibrations. For each nadir spectrum, the most recent

calibration spectra are selected. The data-trimming method is the same as for SWIR. Equation (9) shows how radiance

spectra $L_{b,d}^{obs}$ are generated from earth scene observation complex spectra $S_{b,d}^{obs}$, black body complex spectra $S_{b,d}^{ict}$ and deep

space complex spectra $S_{b,d}^{ds}$:

$$L_{b,d}^{obs} = \left[ \frac{S_{b,d}^{obs} - S_{b,d}^{ds}}{\eta S_{b,d}^{ict} - S_{b,d}^{ds}} - \frac{\varepsilon^{obs} L_{b,d}^{m\_obs} - \varepsilon^{ds} L_{b,d}^{m\_ds}}{(1-\varepsilon^{ds}) \cdot \varepsilon^{ict} \cdot L_{b,d}^{ict} + \varepsilon^{ds} \left( L_{b,d}^{m\_ict} - L_{b,d}^{m\_ds} \right)} \right] \cdot \frac{(1-\varepsilon^{ds}) \cdot \varepsilon^{ict} \cdot L_{b,d}^{ict} + \varepsilon^{ds} \left( L_{b,d}^{m\_ict} - L_{b,d}^{m\_ds} \right)}{(1-\varepsilon^{obs})},$$    (9)

Where,

$L_{b,d}^{ict}$ :    Black body radiance

$L_{b,d}^{m\_obs}$ :    Scan mirror radiance calculated from the mirror temperature measured while viewing an earth scene.

$L_{b,d}^{m\_ds}$ :    Scan mirror radiance calculated from the mirror temperature measured while viewing a deep space scene.

$L_{b,d}^{m\_ict}$ :    Scan mirror radiance calculated from the mirror temperature measured while viewing a black body scene.

$\varepsilon^{obs}$ :    Scan mirror emissivity when viewing an earth scene

$\varepsilon^{ds}$ :    Scan mirror emissivity when viewing a deep space scene

$\varepsilon^{ict}$ :    Black body emissivity

$\eta$ :    Sensitivity correction factor

$N$    :    Sampling points

$L_{b,d}^{ict} \ L_{b,d}^{ds} \ L_{b,d}^{m\_obs} \ L_{b,d}^{m\_ds} \ L_{b,d}^{m\_ict}$ are assumed to follow Planck's law. The prelaunch-characterized black body emissivity

presents higher than 0.999 in an interesting spectral region, so the $1 - \varepsilon^{ict}$ term is not implemented in the current calibration

equation.



Further details of level-1 processing are described in the GOSAT-2 level-1 algorithm theoretical basis document (GOSAT-2 Level-1 *Algorithm Theoretical Basis Document*). Typical radiance spectra for TANSO-FTS-2, recorded on 1

July 2019, are presented in Fig. 8.

## 5 On-orbit performance in the first year of operation

Nominal operations of TANSO-FTS-2 began on 1 February 2019, including periodic instrument calibrations. During the first year of operation, the radiometric, geometric, and spectroscopic parameters were characterized via calibration operations.

The following section describes the characterization method and results. In addition, some calibration parameters (such as the degradation correction factor for the radiances) are updated. The updated products are compared with temporally and spatially coincident observations of the TANSO-FTS on GOSAT, AIRS on AQUA, and IASI on METOP-B.

## 5.1 Operation overview

GOSAT-2 operations have been nominal and continuous during the first year with three short suspensions of observation. On 8 and 24 April 2019, unplanned satellite maneuvers were carried out to avoid collision with space debris. On 11 December 2019, TANSO-FTS-2 was set to safe mode and suspended observation due to an illogical observation plan that was detected on board. The details of suspended periods are given on the GOSAT-2 website (https://prdct.gosat-2.nies.go.jp/en/index.html).

In the initial on-orbit period, it is also important to establish the stability of operations over periodic cycles, with an emphasis on the operational temperature set points. Figure 9 presents the time series of the cooler stage temperatures and the beam splitter temperature in the interferometer as well as the beta angle (sun-satellite angle) and sun-satellite distance. In the early operation period, the temperature of stage 4 was at the set point of 97 K. However, at this temperature setting, the actual temperature shows a time dependency, probably due to unexpected thermal radiance incoming to the  passive

radiation cooler. As a result on 12 April 2019, the temperature control setpoint for stage 4 was changed from 97 K to 100 K. After this set point change, the temperature of stage 4 was controlled well at 100 K without time dependency.

In nominal operation, the radiator of the cooler views deep space and maintains the set point temperatures. During lunar calibration, the flight geometry of the satellite is changed and the view of the radiator is contaminated by Earth's thermal emission. As a result, temperatures of stages 2, 3, and 4 increase rapidly. These perturbed temperatures settle down to the

operational values only six hours after the lunar calibration.

As of 12 July 2019, the beam splitter temperature shifted from 293.5 K to 292.5 K. Thermal analysis with on-orbit data suggested that the temperature gradient around the interferometer is larger than expected. To reduce the temperature gradient, especially at the beam splitter of the interferometer, the controlled set point of the interferometer was changed as





well as the metrology laser control temperature. As a result, the metrology laser wavelength made a small shift at this time.

In summary, in the first year, two of the key instrument temperatures were adjusted to account for on-orbit conditions.

## 5.2 Radiometric characterization

TANSO-FTS-2 has four types of radiometric calibration sources on board: (1) a monochromatic source which accommodates 768 nm and 1543 nm laser diodes with an integrating sphere; (2) a solar diffuser plate made from

Spectralon® equipped with the flip and shutter mechanism; (3) a four-panel black body equipped with three temperature sensors; and (4) a viewing port directed toward deep space. These are called as ILS cal., Sol cal., BB cal., and DS cal., respectively. The ILS cal. has been performed every six days since May 2019 during night and over sea conditions. Solar diffuser calibration is conducted every orbit in the satellite position over Antarctica. Once every three months, the solar diffuser of the primary side is flipped to the back side (secondary side) and exposed to solar light to identify the degradation

of the solar diffuser on the primary side. BB and DS calibrations are periodically scheduled. Nominally, two pairs of BB and DS calibrations are executed on the day side of the orbit and four pairs on the night side. In addition, TANSO-FTS-2 points to the moon once a month during the night side of the orbit and while the pitch maneuver is suspended.

### 5.2.1 SWIR (bands 1, 2, and 3)

Sol cal. and ILS cal. are used to characterize the sensitivity change of the SWIR bands (bands 1, 2, and 3). As mentioned, Sol cal. is conducted in every orbit with the primary solar diffuser. The distance between the sun and satellite positions is taken into account in the normalized signal intensity. In this analysis, the Bidirectional Reflectance Distribution Function (BRDF) of the solar diffuser is not considered. Figure 10 shows the present-time (left panels) and the beta-angle (right panels) dependencies of the signal from the solar diffuser for bands 1p, 1s, 2p, 2s, 3p, and 3s (top to bottom). The

wavenumbers considered for the analysis are listed in Table 5. There were substantial sensitivity changes with time and input angle to the solar diffuser, and these plots suggest that bands 2 and 3 are less time dependent than band 1.

Sol cal. measurements with the "secondary" diffuser plate have been conducted on 15 April, 2 July, 6 October 2019, and on 4 January 2020. Figure 10 (circles) shows that there is no difference between reference and routine plate for the signal level. In other words, the primary diffuser plate shows no degradation.

To make an independent assessment of the time-dependent degradation, the signal intensities of the ILS cal. data were analyzed. Note that the ILS cal. laser sources are expected to operate at constant input intensity over the time period under consideration. The analysis shows a similar time-dependent change and offset on 12 July 2019. The offset is caused by the set point change of the interferometer's interface temperature.





The time-dependent degradation factors for SWIR were determined by assessing Sol cal. data from February 2019 to

March 2020. Equation (10) provides the empirical degradation correction $Y(v, t)$ to be used in equation (8) and table 6 lists the respective parameters $\alpha$, $\beta$, $\gamma$, and $f$:

$$Y(v, t) = \alpha \left( \beta + \gamma \cdot e^{-\frac{t-t_0}{f}} \right),$$ (10)

where $t_0$ is the time on 5 February 2019. The corrected signal levels for Fig. 10 are plotted in Fig.11. Compared to Fig.10, the lines on Fig.11. are indistinguishable, appearing on top of each other. Since the BRDF of the solar diffuser has not been taken into account so far, the signal level still shows a spurious correlation with the input angle. The effect of BRDF is ignored in the earth scene observation.

The calibrated spectral radiances for TANSO-FTS-2 are compared with temporally and spatially coincident TANSO-

FTS spectral radiances. The screening conditions are as follows:

- Less than 2 km between the GOSAT and GOSAT-2 pointing locations

- Less than 80 degrees of the solar zenith angle

- Less than 5 deg difference of viewing angle for GOSAT and GOSAT-2 against pointing location

- Brightness temperature for TIR region is greater than 250 K

- Quality flags: 0

- Less than 10 % of cloud probability assessed by the TANSO-FTS-2 onboard camera

Figure 12 compares the TANSO-FTS and TANSO-FTS-2 radiances for six-day averages. Figure 12 shows that the calibrated TANSO-FTS-2 spectral radiances generally agree with TANSO-FTS spectral radiances within 2 % of the averaged bias and 0.5 % standard deviation. The evaluated numbers are listed in table 7.


### 5.2.2 TIR (bands 4 and 5)

Spectral radiances for the TIR region (bands 4 and 5) are periodically calibrated by applying BB and DS cal. spectra. Nominally, six pairs of these calibrations are done every orbit. To characterize the noise performance of the TIR bands in orbit, the typical Noise-Equivalent differential radiaNce (NEdN) and Noise-Equivalent differential Temperature (NEdT) are

plotted in Fig.13 with the black body temperature trend within a one-revisit time period. Figure 13 upper panel presents the temperature of the black body source which varied by 0.5 K peak-to-peak. To estimate the NEdN and NEdT in orbit, the black body spectra and deep space spectra were processed according to equations (11) and (12) (Chen et al., 2015),

$$NEdN_{b,d}^{ict}[\sigma] = Re \left\{ stdev \left\{ L_{b,d}^{ict\_obs}[i, \sigma] \right\} \right\}$$ (11)




$$L_{b,d}^{ict\_obs}[i,\sigma] = \left( \frac{S_{b,d}^{ict}[i,\sigma] - mean\langle S_{b,d}^{ds}[i,\sigma] \rangle}{mean\langle S_{b,d}^{ict}[i,\sigma] \rangle - mean\langle S_{b,d}^{ds}[i,\sigma] \rangle} \right) \cdot L_{b,d}^{ict}[i,\sigma] \tag{12}$$

where $i$ is the $i$-th measurement and the other symbols are the same as defined above for equation (9). Typically, there are up to 24 calibration measurements $i$ per orbit. Figure 13 shows that NEdNs for both bands 4 and 5 have almost the same

noise level. NEdT is less than 0.3 K against the typical black body temperature condition (around 294.2 K).

TIR V102102 calibrated spectra are compared with other satellite data, which are AIRS on AQUA and IASI on METOP-B. The coincident latitudes between AIRS and TANSO-FTS-2 and between IASI and TANSO-FTS-2 are illustrated in Fig. 14. The coincident points between AIRS and TANSO-FTS-2 are located in the mid-latitudes, and those of IASI and TANSO-FTS-2 are located at high latitudes. These conditions lead to comparison with different brightness temperature

ranges. We focused on the comparison in the following spectral regions: $CO_2$ channel (681.99-691.66 cm$^{-1}$), window channel (900.31-903.78 cm$^{-1}$), ozone ($O_3$) channel (1030.08-1039.69 cm$^{-1}$), and $CH_4$ channel (1304.36-1306.68 cm$^{-1}$). The coincidence criteria are as follows;

- Less than +/-100 km and +/-5 min between GOSAT-2 and AIRS/IASI orbit

- Smaller than +/-3 degree of cross-track and +/-3 degree of along-track angle for GOSAT-2

- Less than 17 km between GOSAT-2 and AIRS/IASI observing locations

Since the spectral resolution of AIRS is lower than that of TANSO-FTS-2, we convolve the TANSO-FTS-2 spectra with the AIRS spectral response function. The same scheme is applied to IASI data. The inter-comparison method is the same as the one used by Kataoka (2019).

Figure 15 shows the comparison results. The mean temperature differences between TANSO-FTS-2 and AIRS, and

TANSO-FTS-2 and IASI are less than 0.5 % from February 2019 to March 2020. In the case of TANSO-FTS-2 and AIRS comparison, the coincident scenes have moderate brightness temperatures such as 280 K. In contrast, for the TANSO-FTS-2 and IASI comparison, the brightness temperature is colder than that for the AIRS case. Figure 15 (b) shows the differences at four focused channels against the window temperature. This figure suggests that the differences are almost zero over the 240K window temperature. However, it has 1 to 2 K bias in the cold scene with some wavenumber dependencies.

Figures 15 (c) and (d) present the time series of the brightness temperatures difference between TANSO-FTS-2 and IASI and between TANSO-FTS-2 and AIRS, showing that cold scene targets such as the $CO_2$ and $CH_4$ channels exhibit a 1 to 2 K bias. However, the bias is less than 0.5 K in the warm target case.

### 5.2.3 Spectral radiance characterization challenges

In the first year of operation, the spectral radiances for both the SWIR and TIR were recalibrated and the radiance conversion

coefficients were updated. In parallel, several challenges were identified, for example, the SWIR polarization sensitivity has changed, and the dependence of TIR calibration on scene temperature and wavenumber changed.





As described in the previous section, TANSO-FTS-2 shows considerable sensitivity to the U Stokes vector component. The polarization sensitivity has been characterized on the ground by positioning a linear polarizer in front of the instrument. In contrast, on-orbit data suggests that these polarization sensitives changed after launch. A future update of the level-1
product will include the best estimated radiances and the related polarization model. For updating the polarization model, we require feedback from retrieval studies that examine retrieval performance for polarizing scenes.

The correction term $\eta$ in equation (9) for calibrating the TIR radiance is ideally equal to 1, but empirically differs from 1 to compensate for the bias. As a result, the spectral radiance differences between TANSO-FTS-2 and AIRS/IASI are minimized, and the brightness temperature difference is less than 0.5 % during the first year when the sensitivity correction
factor $\eta$ is set to 1.0198. However, $\eta$ is estimated by an empirical method. Currently, we assume that $\eta$ deviating from 1 originates from unaccounted polarization sensitivity in the TIR optics since the input polarized light geometry is completely different between nadir observations and black body observations. In addition, $\eta$ might have wavenumber dependence. For future versions of level-1 products, a theoretical model will be constructed and implemented for level-1 processing.

**5.3 Geometric characterization**

To identify a line-of-sight offset, the processed pointing locations based on satellite position and the along-track (AT) and cross-track (CT) pointing angles are compared with the validated ground control position based on co-registered camera images. The time series of the root-mean-square differences of latitude and longitude between the processed pointing locations and the validated locations are plotted in Fig.16 for the period 5 February 2019 to 31 March 2020. Figure 16 (a)
shows that there is no time dependence in the differences. Likewise, the latitude and longitude differences are plotted in Fig. 16 (b). The graph shows that TANSO-FTS-2 points have almost no offset. The averaged differences are less than 0.02 km in latitude and 0.06 km in longitude. The standard deviation of differences is 0.17 km in latitude and 0.18 km in longitude.

**5.4 Spectroscopic characterization**

Through the match-up analysis between TANSO-FTS and TANSO-FTS-2, we found that the absorption spectra of TANSO-FTS-2 show a marginally coarser spectral resolution than TANSO-FTS in bands 1 to 3. Theoretically, the spectral resolution between TANSO-FTS and TANSO-FTS-2 should be the same. However, optical aberration and alignment is slightly different between the two instruments. During prelaunch characterization, the ILS function was derived from monochromatic light source measurements. Theoretically, the response to monochromatic light can provide the proper ILS given that the
light beam is uniform and covers the full FOV. Non-uniformity of the light beam or partial illumination of the detector can lead to a narrower line shape. Therefore, the ILS function is reassessed with on-orbit data.



### 5.4.1 In-orbit ILS calibration

The designed spectral resolution is 0.2cm$^{-1}$ for all bands. Due to the finite field of view, optical aberration, and
misalignment, the theoretical sinc-function is distorted and spectral resolution is worse.

TANSO-FTS-2 accommodates monochromatic light sources. These light sources allow us to monitor the changes in the
ILS in orbit. Typically, ILS at shorter wavelengths are more susceptible to alignment or illumination changes. The 0.77 um
laser diode is preferred for identifying the changes.

Figure 17 presents the ILS function for bands 1 and 2 based on the laser diode signal, and the trend of the FWHM (full-
width-at-half-maximum) for both wavelengths. To create densely sampled spectra, we applied zero-padding to the original
OPD-sampled interferograms and retrieved the ILS with higher sampling. The shapes of bands 1 and 2 are significantly
different. Band 1 is the most sensitive to optical alignment and has an asymmetric line shape. In contrast, band 2 has an
almost symmetric line shape.

### 5.4.2 Spectral response characterization challenges

As described in the previous section, ILSs show a time dependence. This might be due to optical alignment change with
time. Not only the ILS function, but also polarization sensitivities were found to change over time because the ratio between
p and s polarization signal was changing. The first year of operation data suggest that the rate of change is mild and was
likely to become progressively smaller. Since July 2019, the instrumental line shape is almost constant.

For level-1 version102102 auxiliary data, a best-estimate ILS function is provided on the ILS cal. and Sol cal. datasets.
The best-estimate ILS function is slightly wider than that of the prelaunch testing. However, a time-dependent term is not
implemented. The typical residual between observed spectra and theoretical spectra is reduced with the current best-estimate
ILS function and plotted in Fig.18. In a future update, the time-dependent ILS function will have to be improved; it is
especially needed for Band 1.


### 5.5 Intelligent pointing functionality

TANSO-FTS-2 carries a camera which can identify cloudy areas in the TANSO-FTS-2 field of view before observation.
Based on onboard processing of the camera images, TANSO-FTS-2 relocates the observation point to a cloud-free area.
Since the processing resources in orbit are limited, a simple and fast cloud identification algorithm was implemented.




### 5.5.1 Cloud identification algorithm

A camera image pixel is decomposed into three-pixel information (red, green, and blue). A simple cloud-detection algorithm uses these raw three-pixel measures to directly identify the cloud-contaminated pixels in the image. The following three indicators are used


$$S = max(R, G, B) \tag{13}$$

$$M = min(R, G, B) \tag{14}$$

$\quad V = 255 * \frac{S-M}{S} \tag{15}$

To identify the most probable cloud locations, the following filters are applied with thresholds of $a$, $b$, $c$, $d$, $e$, and $f$ equal to 73, 16, 83, 2, 4, and 60, respectively.

$\quad Filter\ 1 : S > a\ \&\ V < b \tag{16}$

$$Filter\ 2 : M > c \tag{17}$$

$$Filter\ 3 : abs(R - G) \le d\ \&\ B - \frac{(G+R)}{2} > e \tag{18}$$


$$Filter\ 4 : M < f \tag{19}$$

If the above conditions for filters 1 or 2 or 3 conditions are confirmed, and those for filter 4 are not, the scene is identified as probably cloud covered.

The result of cloud detection for each scene on orbit is not recorded. However, we did add auxiliary information about onboard cloud identification by applying the same algorithms to the camera images during on-ground processing. Using this information allows on-orbit performance to be evaluated. To this end, for each TANSO-FTS-2 measurement, we take the camera images taken before and after each FTS observation and run the above cloud-detection algorithm on the images. We calculate a cloud cover index for each FTS observation by finding the ratio of cloudy camera image pixels to the total

number of camera pixels within the FTS footprint. Thus, the cloud cover index is 0 for cloudless scenes, and the index is 100 % for fully cloudy scenes.

Figure 19 displays a typical global map for cloud cover index lower than 1 % for September 2019, comparing the cloud cover index before (upper panel) and after (lower panel) intelligent pointing. Clear differences are observed over central





America, the Amazon, central Africa, and southeast Asia. In these areas, with intelligent pointing, the number of
measurements with a low cloud cover index is increased by 1.7, 1.6, 1.9 and 2.1, respectively. Globally, for the study period
from March to December 2019, the number of clear-sky retrievals (cloud cover index up to 1%) was increased by a factor of
1.8 over land for intelligent pointing over standard pointing.

### 5.5.2 Intelligent pointing challenges

The cloud detection algorithm for intelligent pointing is based on simple brightness and chroma thresholds. For darker
scenes, such as ponds and lakes, the method tends to fail to detect cloudy areas. For brighter surfaces, such as concrete
buildings in a city, the method has too many cloud detections. Both false positive and false negative detections imply the risk
for unsuitable re-pointing operations for the FTS, so it might be better to examine the filter thresholds and the observation
plan region by region. If we identify areas on the globe that are unsuitable for re-pointing, intelligent pointing can be
switched off and on depending on the location. In the current operation, intelligent pointing is switched off over Cal/Val.
sites and  user-specified sites.

## 6. Conclusions

The Japanese Greenhouse gases Observing SATellite-2 (GOSAT-2), in orbit since October 2018, is the follow-up mission of
GOSAT, which has been operating since January 2009. Both satellites are dedicated to the monitoring of global carbon
dioxide and methane to further knowledge of the global carbon cycle. This paper has reported on the function and
performance of the TANSO-FTS-2 instrument, level-1 data processing, and calibrations for the first year of GOSAT-2
observation. To evaluate its performance, the spectral radiances (level-1 processor version v102102) collected by TANSO-
FTS-2 between February 2019 and March 2020 are compared with the spatiotemporally coincident measurements of the
TANSO-FTS on GOSAT for the SWIR band, and with AIRS on AQUA, IASI on METOP-B for the TIR bands. We
conclude that the spectral radiances measured by TANSO-FTS and TANSO-FTS-2 agree to within 2 % for the SWIR bands.
In the TIR, the agreement between TANSO-FTS-2 and AIRS, IASI is better than 0.5 % (1 K) for scenes brighter than 220 K.
We further evaluated GOSAT-2's intelligent pointing mechanism based on active cloud avoidance. The preliminary analysis
indicates that the number of scenes useful for spectral analysis increased by factor 1.8 over a stiff pointing schedule.

*Data availability.*

All datasets used here are publicly available and can be accessed through the links and references provided.



*Author contributions.*

HS wrote the manuscript and performed the data analysis with support from FK, NK, RO and AB. RO, AB, MH, HB and AK contributed to the interpretation of the results. FK, RO, KS, and HI supported to the satellite inter-comparison data preparation or expertise on data sets. All authors discussed the results and contributed to the manuscript.

*Competing interests.*

The authors declare that they have no conflict of interest.

*Acknowledgments.*

The authors would like to thank J. Kasuya, Y. Ito of the Mitsubishi Electronics Corporation, Y. Yata of the Mitsubishi Space Software Corporation, T. Kaku of the Remote Sensing Technology Center, and M. Buchwitz of the University of Bremen for
their useful  suggestions, as well as  the members of the Japanese Ministry of the Environment, the National Institute for Environmental Studies, L3 Harris, ABB Inc. for their cooperation.





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





**Tables:**


**Table 1.** Orbit parameters of GOSAT-2 and GOSAT.

| | Specification Items | GOSAT-2 Specifications | GOSAT-2 Remarks | GOSAT Specifications | GOSAT Remarks |
|---|---|---|---|---|---|
| | Type | Sun synchronous, quasi-recurrent | | Sun synchronous, quasi-recurrent | |
| | Local overpass time | 13hours±15minutes | Descending node | 13hours±15minutes | Descending node |
| | Altitude | 612.98km | Not including altitude variations in orbit | 666 ± 0.6km | |
| O R B I T | Inclination angle | 97.84degrees | | 98.0degrees ± 0.1degrees | |
| | Eccentricity | 0.00106 | Frozen orbit | | Frozen orbit |
| | Period | Approximately 98.1minutes | | Approximately 98.1minutes | |
| | Repeat cycle | 6days (89paths) | | 3days (44paths) | |
| | Origin point | An orbit exactly over Lamont, OK (Latitude 36.6North, Longitude 97.5 West) | | An orbit exactly over Tsukuba, Ibaraki (Latitude 36.1North, Longitude 140.1 East) | |
| | Descending node accuracy | ±2.5km | Depending upon the frequency of orbit control manuevers | ±2.5km | Depending upon the frequency of orbit control manuevers |




**Table 2.** Spectroscopic specifications of the TANSO-FTS-2.

|  | Band 1 | Band 2 | Band 3 | Band 4 | Band 5 |
|---|---|---|---|---|---|
| Spectral Coverage (cm$^{-1}$) | 12950-13250 | 5900-6400 | 4200-5200 | 1188-1800 | 700-1188 |
| Polarization Obs. | 2 | 2 | 2 | N/A | N/A |
| Spectral Resolution (cm$^{-1}$) | 0.2 cm$^{-1}$(Both sides scan) (MOPD +/-2.5 cm) | | | | |
| Sampling Number | 153090 | 76545 | 76545 | 38250 | 38250 |
| Full Width Half Maximum (cm$^{-1}$) | < 0.4 | < 0.27 | < 0.27 | < 0.27 | < 0.27 |
| Detector | Si | PV-MCT | PV-MCT | PV-MCT | PC-MCT |




**Table 3.** Fitting parameters of on-orbit SNR model.

|   | Band 1p | Band 1s | Band 2p | Band 2s | Band 3p | Band 3s | Band 4 | Band 5 |
|---|---------|---------|---------|---------|---------|---------|--------|--------|
| a | 1.68e-9 | 0.96e-9 | 0.65e-9 | 0.56e-9 | 0.42e-9 | 0.30e-9 | 1.65e-9 | 1.06e-9 |
| b | 3.19e-6 | 2.37e-6 | 1.87e-6 | 1.49e-6 | 1.34e-6 | 1.02e-9 | 1.73e-6 | 8.37e-7 |
| c | 0 | 0 | 0 | 0 | 0 | 0 | 1.70e-6 | 1.59e-6 |





**Table 4.** In-band and out-of-band regions.

| band | In-band region | Out-of-band region | |
|------|----------------|--------------------|--|
| | $[\text{cm}^{-1}]$ | Lower $[\text{cm}^{-1}]$ | Higher $[\text{cm}^{-1}]$ |
| 1 | 12950 − 13250 | 12450 − 12550 | 13650 − 13750 |
| 2 | 5900 − 6400 | 4800 − 4900 | 7000 − 7100 |
| 3 | 4200 − 5200 | 3800 − 3900 | 5700 − 5800 |
| 4 | 1188 − 1800 | 988 - 1038 | 1950 − 2000 |
| 5 | 700 − 118 | 500 − 600 | 1288 − 1388 |






**Table 5.** Wavenumbers considered for the radiometric performance analyses.

| Color | Band 1p | Band 1s | Band 2p | Band 2s | Band 3p | Band 3s |
|---|---|---|---|---|---|---|
| red | 12975.2 cm$^{-1}$ | 12975.2 cm$^{-1}$ | 6174.7 cm$^{-1}$ | 6174.7 cm$^{-1}$ | 4808.8 cm$^{-1}$ | 4808.8 cm$^{-1}$ |
| orange | 12993.8 cm$^{-1}$ | 12993.8 cm$^{-1}$ | 6186.8 cm$^{-1}$ | 6186.8 cm$^{-1}$ | 4822.9 cm$^{-1}$ | 4822.9 cm$^{-1}$ |
| green | 13027.6 cm$^{-1}$ | 13027.6 cm$^{-1}$ | 6229.6 cm$^{-1}$ | 6229.6 cm$^{-1}$ | 4849.9 cm$^{-1}$ | 4849.9 cm$^{-1}$ |
| blue | 13122.1 cm$^{-1}$ | 13122.1 cm$^{-1}$ | 6257.8 cm$^{-1}$ | 6257.8 cm$^{-1}$ | 4871.4 cm$^{-1}$ | 4871.4 cm$^{-1}$ |
| purple | 13171.8 cm$^{-1}$ | 13171.8 cm$^{-1}$ | 6277.9 cm$^{-1}$ | 6277.9 cm$^{-1}$ | 4880.4 cm$^{-1}$ | 4880.4 cm$^{-1}$ |






**Table 6.** Radiometric correction parameters as function of time since launch

| | Band 1p | | Band 1s | | Band 2p | | Band 2s | | Band 3p | | Band 3s | |
|---|---|---|---|---|---|---|---|---|---|---|---|---|
| | 1[*] | 2[**] | 1 | 2 | 1 | 2 | 1 | 2 | 1 | 2 | 1 | 2 |
| $\alpha$ | 1 | 1 | 1 | 1 | 1 | 0.993 | 1 | 0.993 | 1 | 0.976 | 1 | 0.976 |
| $\beta$ | 0.7557 | 0.6225 | 0.7809 | 0.6995 | 1 | 1 | 1 | 1 | 0.9797 | 1 | 0.9797 | 1 |
| $\gamma$ | 0.2113 | 0.1541 | 0.2191 | 0.0922 | 0 | 0 | 0 | 0 | 0.0236 | 0 | 0.02 | 0 |
| $f$ | 68.019 | 656.80 | 66.855 | 654.83 | 1 | 1 | 1 | 1 | 79.635 | 1 | 79.635 | 1 |
| $t_0$ | | | | | | 2019/02/05 | | | | | | |

[*] Time period: 2019/02/05 – 2019/07/12.

[**] Time period: 2019/07/13 – .




**Table 7.** Radiance comparison between TANSO-FTS and TANSO-FTS-2 in SWIR region

| Band 1p | | Band 1s | | Band 2p | | Band 2s | | Band 3p | | Band 3s | |
|---|---|---|---|---|---|---|---|---|---|---|---|
| Ave. | Std. | Ave. | Std. | Ave. | Std. | Ave. | Std. | Ave. | Std. | Ave. | Std. |
| 12975.2 cm$^{-1}$ | | 12975.2 cm$^{-1}$ | | 6174.7 cm$^{-1}$ | | 6174.7 cm$^{-1}$ | | 4808.8 cm$^{-1}$ | | 4808.8 cm$^{-1}$ | |
| 1.011 | 0.314 | 0.987 | 0.325 | 1.000 | 0.376 | 1.01 | 0.332 | 1.021 | 0.479 | 1.022 | 0.48 |
| 12993.8 cm$^{-1}$ | | 12993.8 cm$^{-1}$ | | 6186.8 cm$^{-1}$ | | 6186.8 cm$^{-1}$ | | 4822.9 cm$^{-1}$ | | 4822.9 cm$^{-1}$ | |
| 1.013 | 0.315 | 0.995 | 0.328 | 1.003 | 0.377 | 1.01 | 0.333 | 1.03 | 0.484 | 1.033 | 0.485 |
| 13027.6 cm$^{-1}$ | | 13027.6 cm$^{-1}$ | | 6229.6 cm$^{-1}$ | | 6229.6 cm$^{-1}$ | | 4849.9 cm$^{-1}$ | | 4849.9 cm$^{-1}$ | |
| 1.001 | 0.311 | 0.994 | 0.327 | 1.000 | 0.376 | 1.003 | 0.33 | 1.015 | 0.477 | 1.021 | 0.479 |
| 13122.1 cm$^{-1}$ | | 13122.1 cm$^{-1}$ | | 6257.8 cm$^{-1}$ | | 6257.8 cm$^{-1}$ | | 4871.4 cm$^{-1}$ | | 4871.4 cm$^{-1}$ | |
| 0.984 | 0.306 | 0.982 | 0.323 | 0.997 | 0.375 | 0.995 | 0.327 | 0.987 | 0.464 | 0.995 | 0.467 |
| 13171.8 cm$^{-1}$ | | 13171.8 cm$^{-1}$ | | 6277.9 cm$^{-1}$ | | 6277.9 cm$^{-1}$ | | 4880.4 cm$^{-1}$ | | 4880.4 cm$^{-1}$ | |
| 0.985 | 0.306 | 0.976 | 0.322 | 1.001 | 0.376 | 0.997 | 0.328 | 1.016 | 0.477 | 1.021 | 0.479 |
| Total | | | | | | | | | | | |
| 0.999 | 0.31 | 0.987 | 0.325 | 1.000 | 0.376 | 1.003 | 0.33 | 1.014 | 0.476 | 1.018 | 0.478 |






**Figures:**


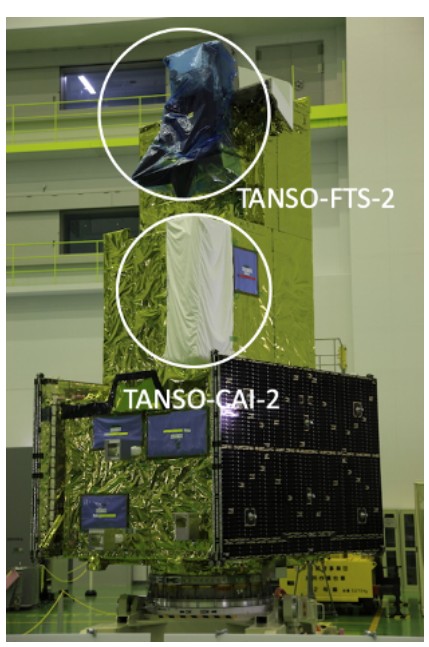

**Figure 1.** Photograph of GOSAT-2 before launch.







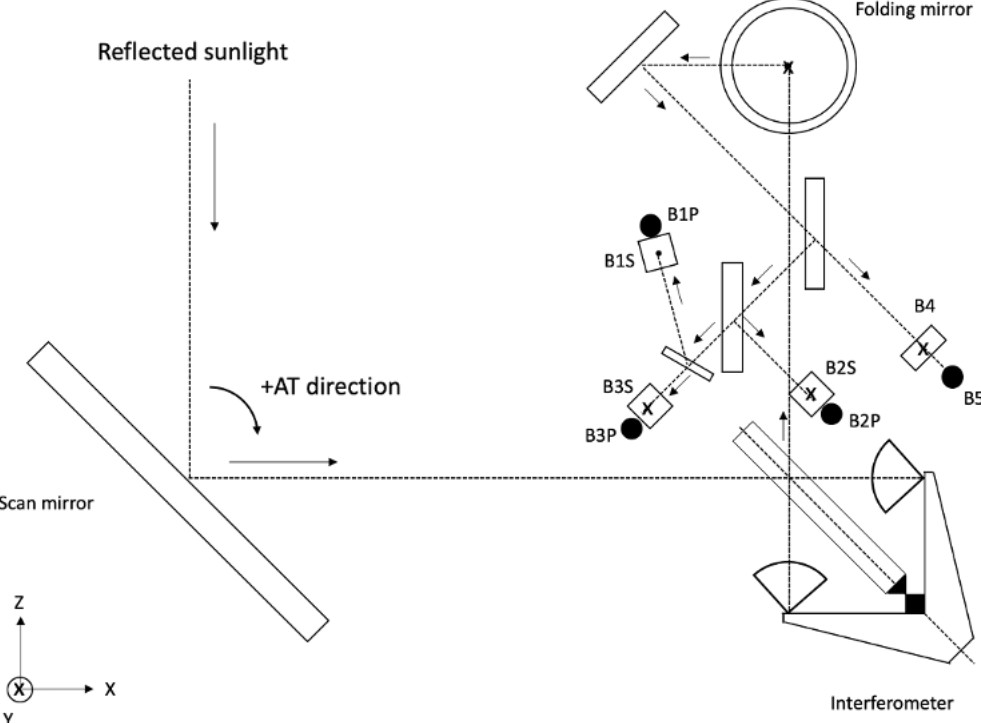

**Figure 2.** Schematic diagram for TANSO-FTS-2 optical layout.







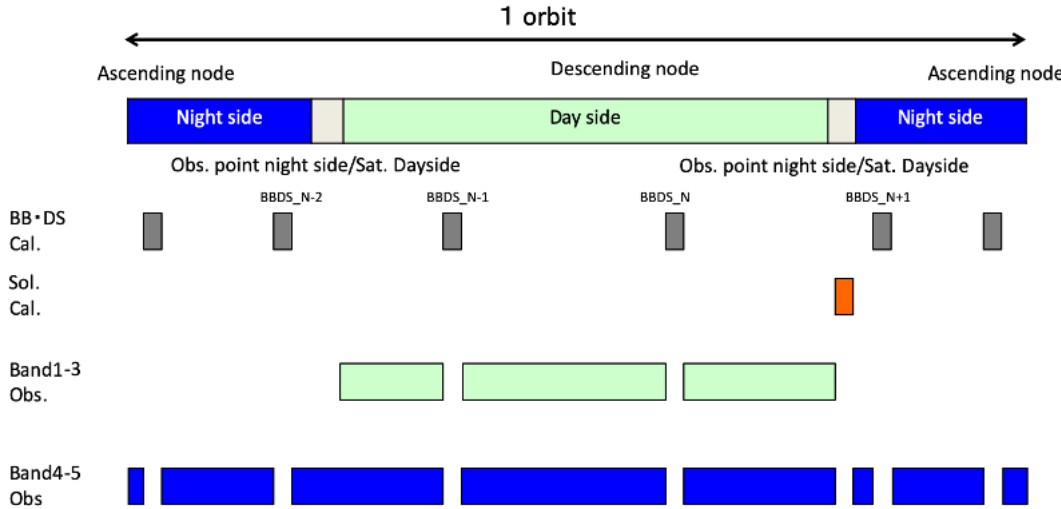


**Figure 3.** Typical calibration operations during an orbit.






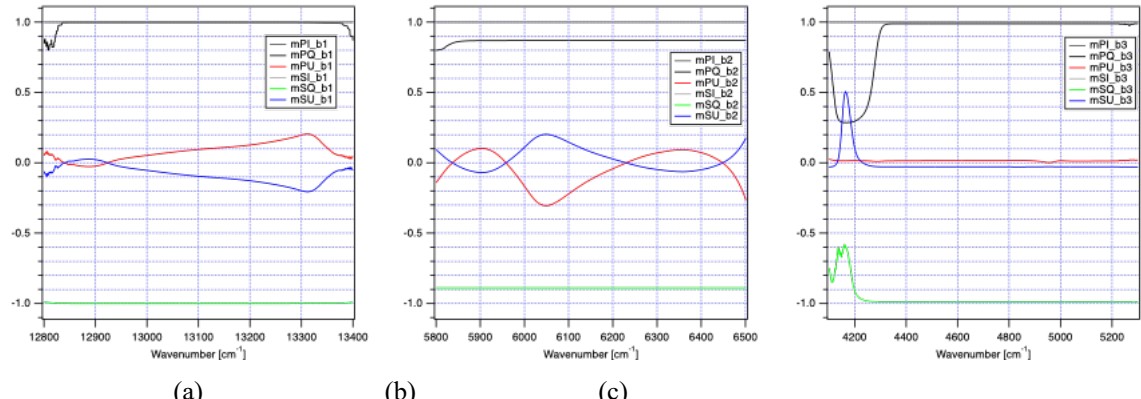

           (a)             (b)           (c)

**Figure 4.** The estimated Mueller matrix coefficients against I, Q, and U components: (**a**) Band 1; (**b**) Band 2; (**c**) Band 3.







**Figure 5.** Signal to noise calculated from on-orbit data: (**a**) p-polarization bands for band 1,2,3, (**b**) s-polarization band for band 1,2,3, (**c**) band 4, and (**d**) band 5. Black dots show the model lines.








(a)

(b)

(c)

**Figure 6.** The simulated instrument line shape function (ILS): (**a**) p-polarization bands for SWIR; (**b**) s-polarization bands for SWIR; (**c**) TIR bands.





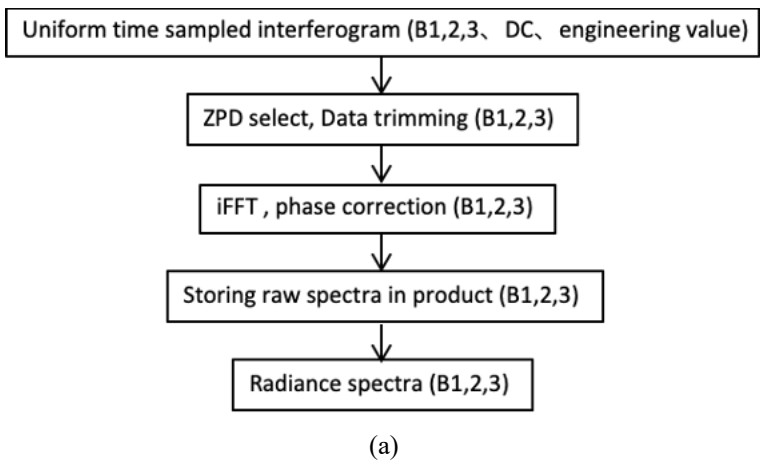

(a)


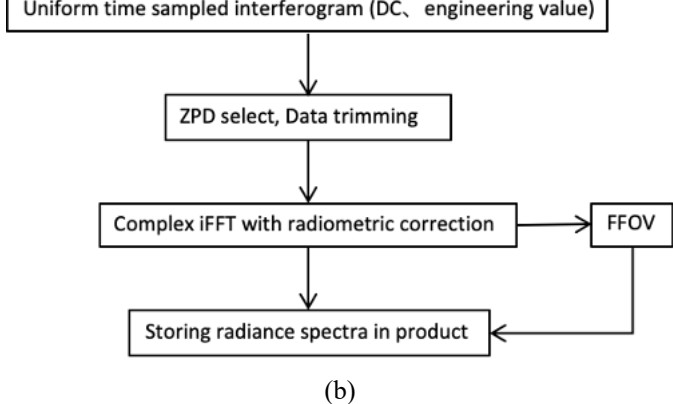

(b)

**Figure 7.** Processing flow for SWIR bands (**a**) and for TIR bands (**b**).





**Figure 8.** Typical spectra for TANSO-FTS-2 over a vicarious calibration site (Railroad Valley, USA).






**Figure 9.** Time series of temperatures (**a**)cooler stage1, (**b**)cooler stage2, (**c**)cooler stage3, (**d**)cooler stage4, (**e**)beam splitter on interferometer, (**f**) satellite beta-angle (grey with left axis) and (**g**)sun-satellite distance (black with right axis).





**Figure 10.** Time and beta-angle dependent diffused solar signal level for SWIR region: left panels; time-dependency, right panel; beta-angle dependency.






**Figure 11.** Compensated diffused solar signal level for SWIR region: left panels; time-dependency, right panel; beta-angle dependency.






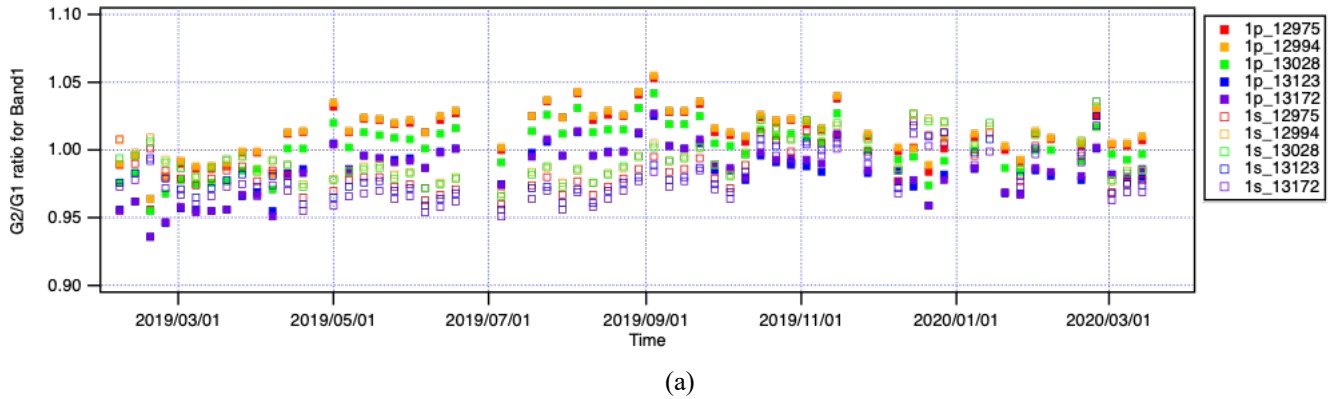

(a)


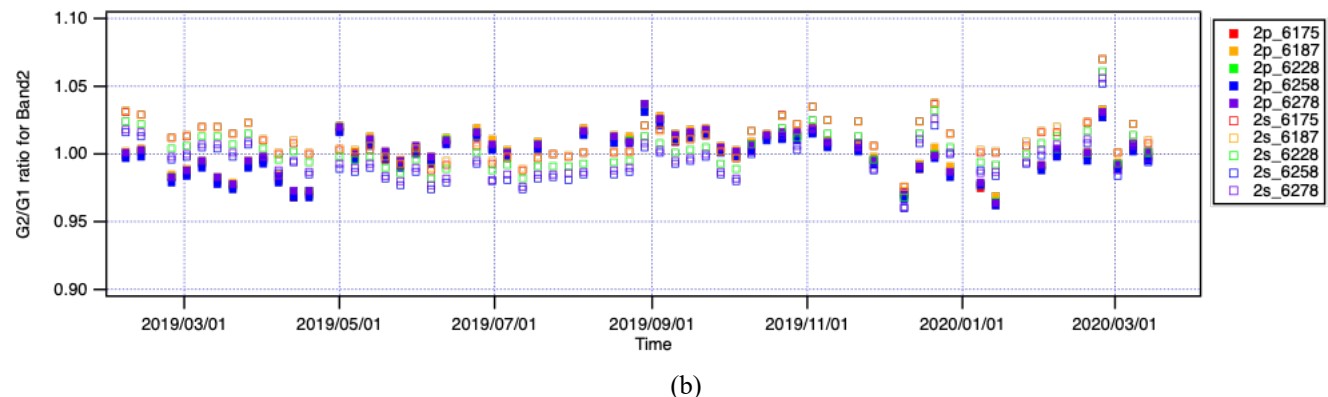

(b)

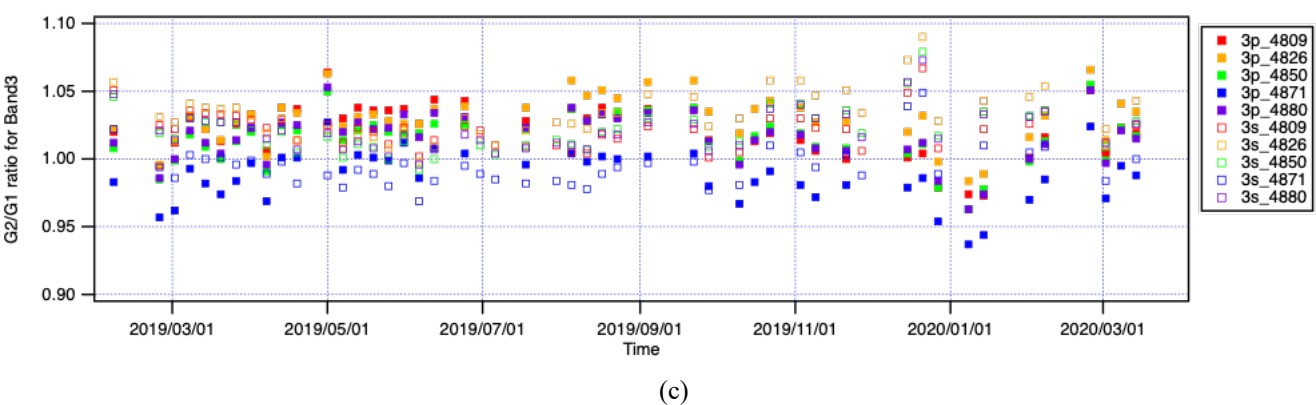

(c)

**Figure 12.** Inter comparison of radiance spectra between TANSO-FTS onboard GOSAT (G1) and TANSO-FTS-2 onboard GOSAT-2 (G2): (**a**) Band 1; (**b**) Band 2; (**c**) Band 3.




**Figure 13.** Noise characteristics of TIR bands 4 and 5: (**a**) typical time series of black body temperature, (**b**) noise

equivalent differential radiance (NEdN), (**c**) noise equivalent differential temperature (NEdT).

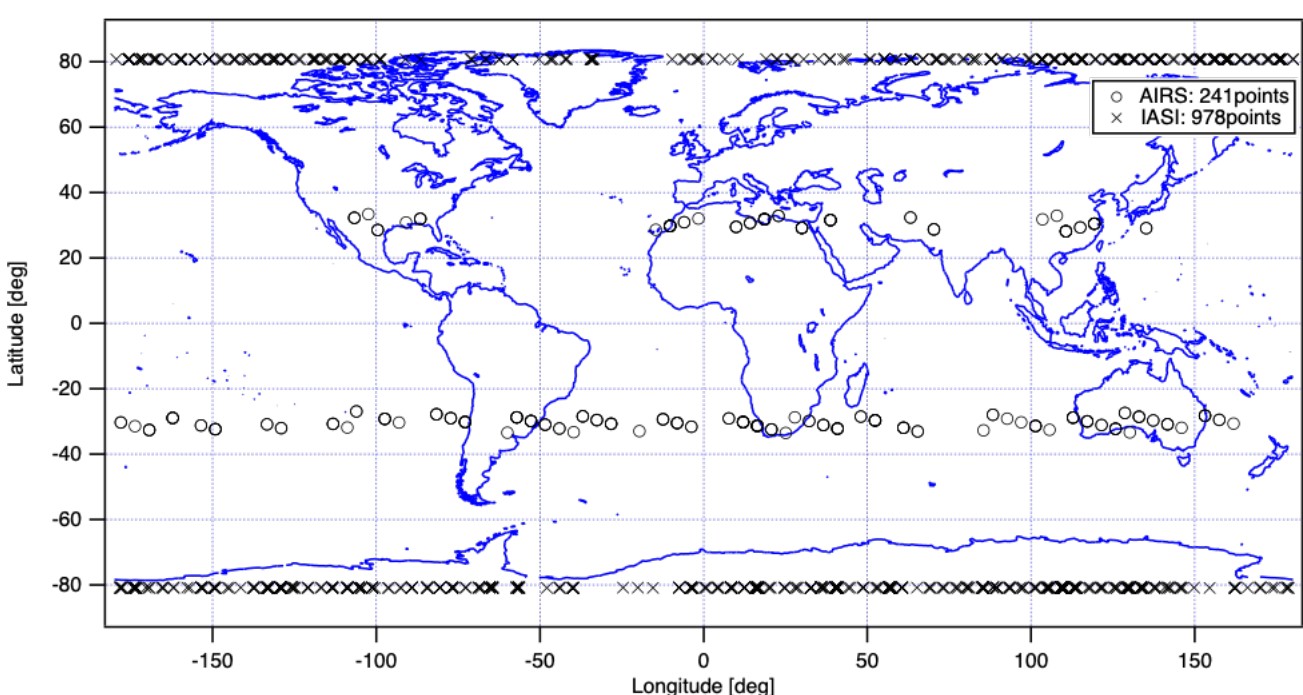


**Figure 14.** Inter comparison between TANSO-FTS-2 and others; coincident latitude and longitude map between TANSO-FTS-2, AQUA (o) and IASI (x).









(a)

(b)



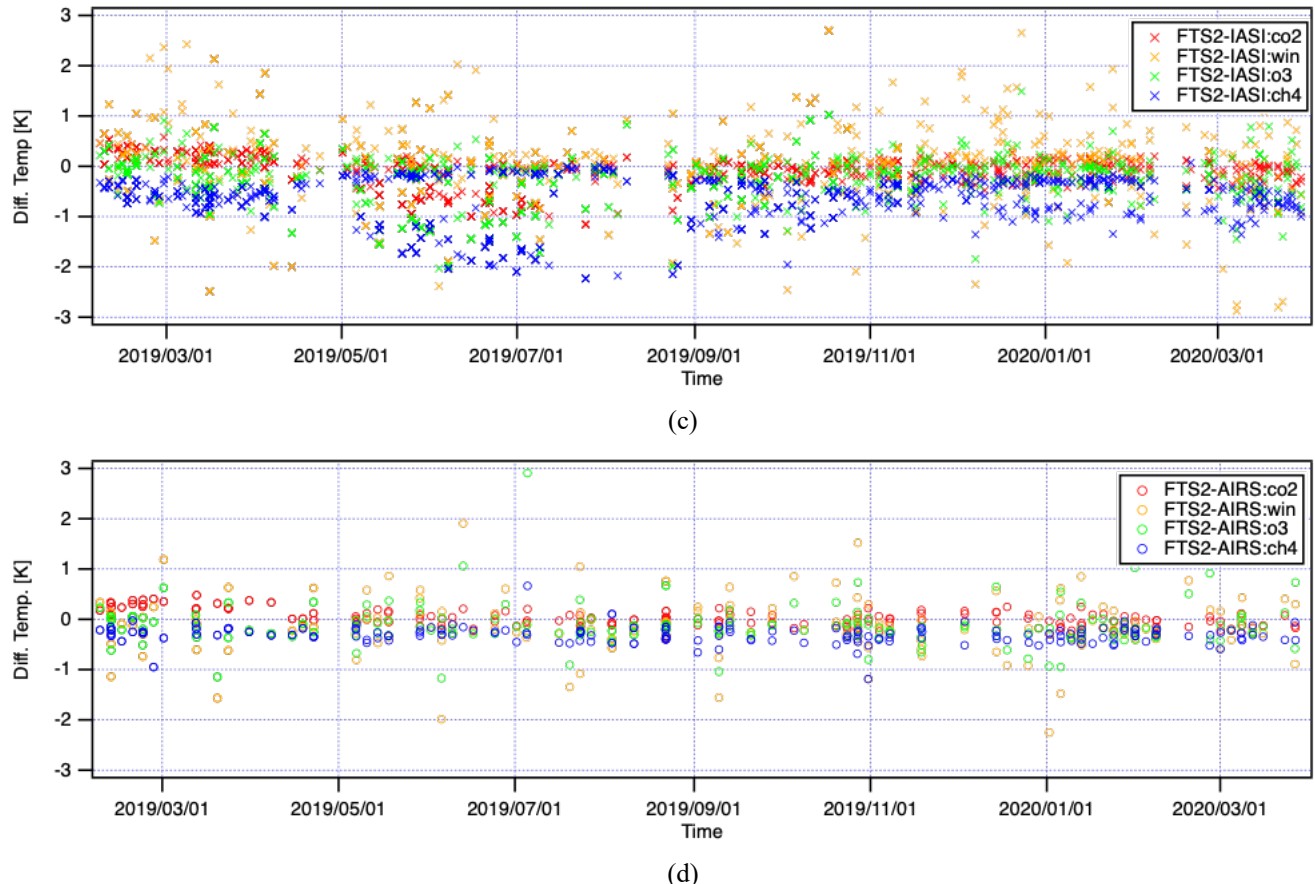

(c)

(d)


**Figure 15.** Inter comparison between TANSO-FTS-2, IASI and AIRS; (**a**) entire averaged difference brightness temperature between TANSO-FTS-2, AIRS and IASI in the entire spectral range; (**b**) channel dependent difference against window

temperature; (**c**) time series of averaged difference brightness temperature between TANSO-FTS-2 and IASI within spectral regions of $CO_2$, window, $O_3$, and $CH_4$ channels; (**d**) the same as (**c**) except for TANSO-FTS-2 and AIRS.



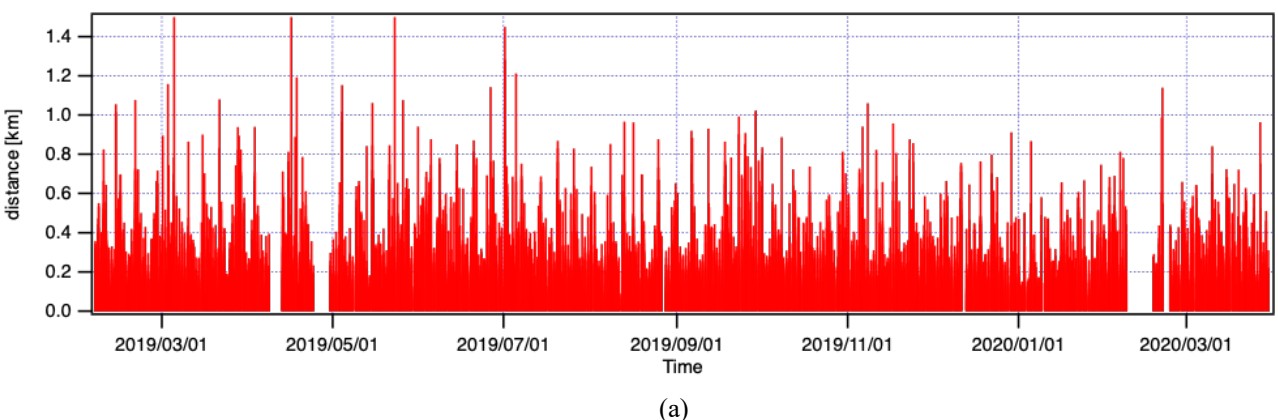

(a)

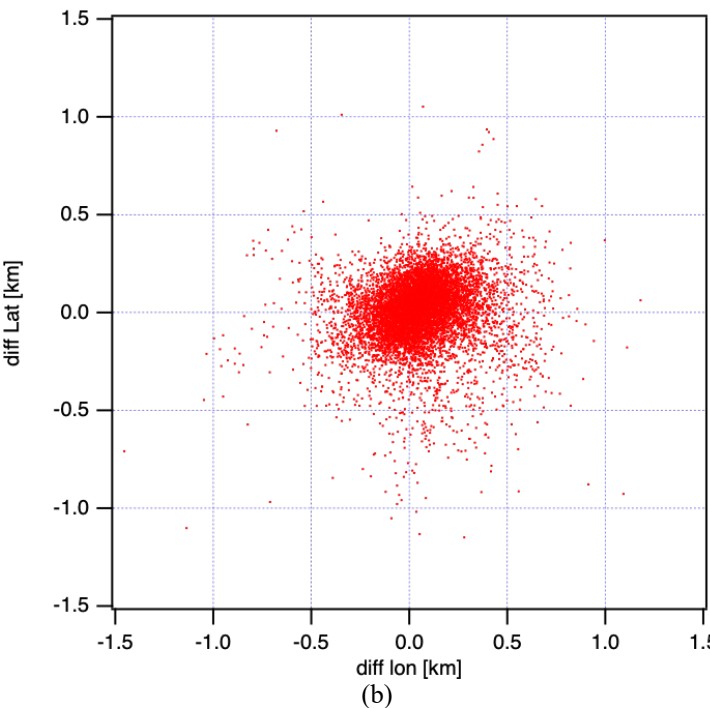

(b)

**Figure 16.** Geometric difference between the processed pointing location and the ground control position using onboard camera images;(**a**) time series since launch; (**b**) Scatter plot in latitude and longitude.






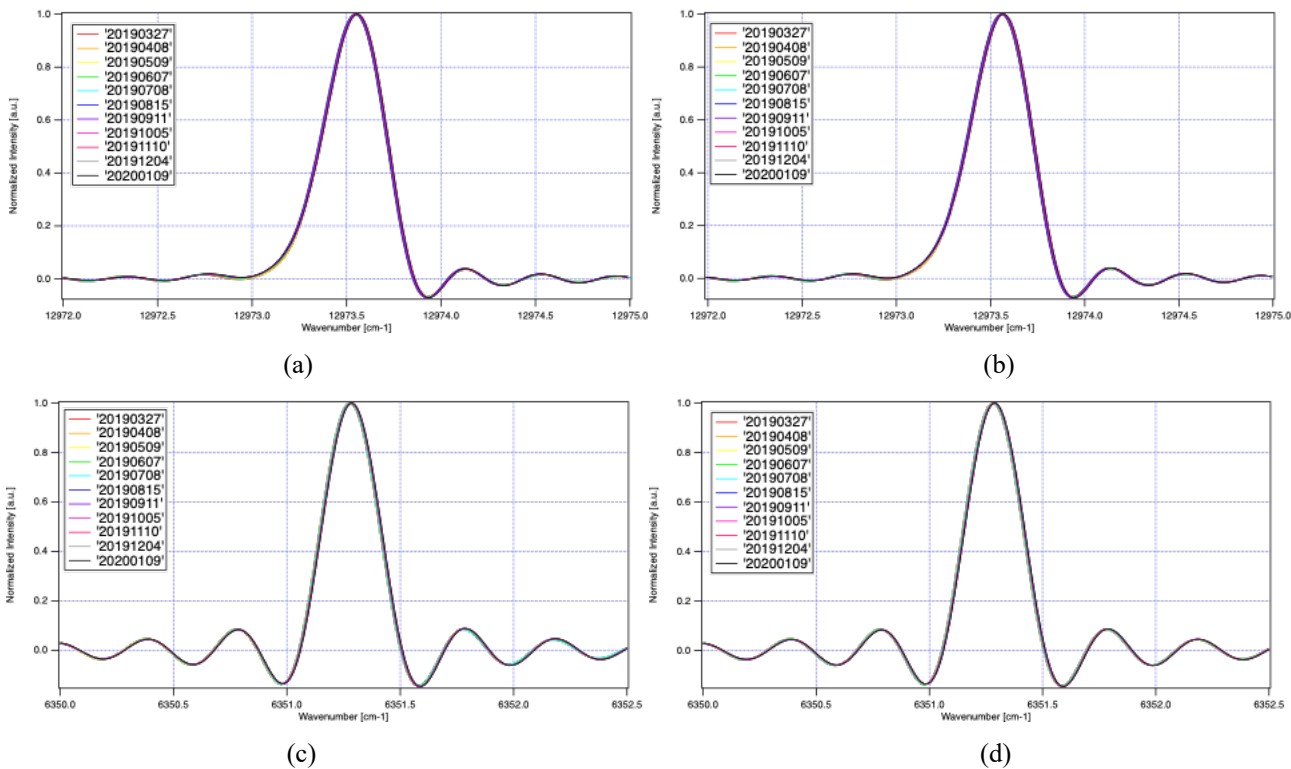

**Figure 17.** On orbit instrument line shape functions; (**a**) Band 1p; (**b**) Band 1s; (**c**) Band 2p; (**d**) Band 2s.




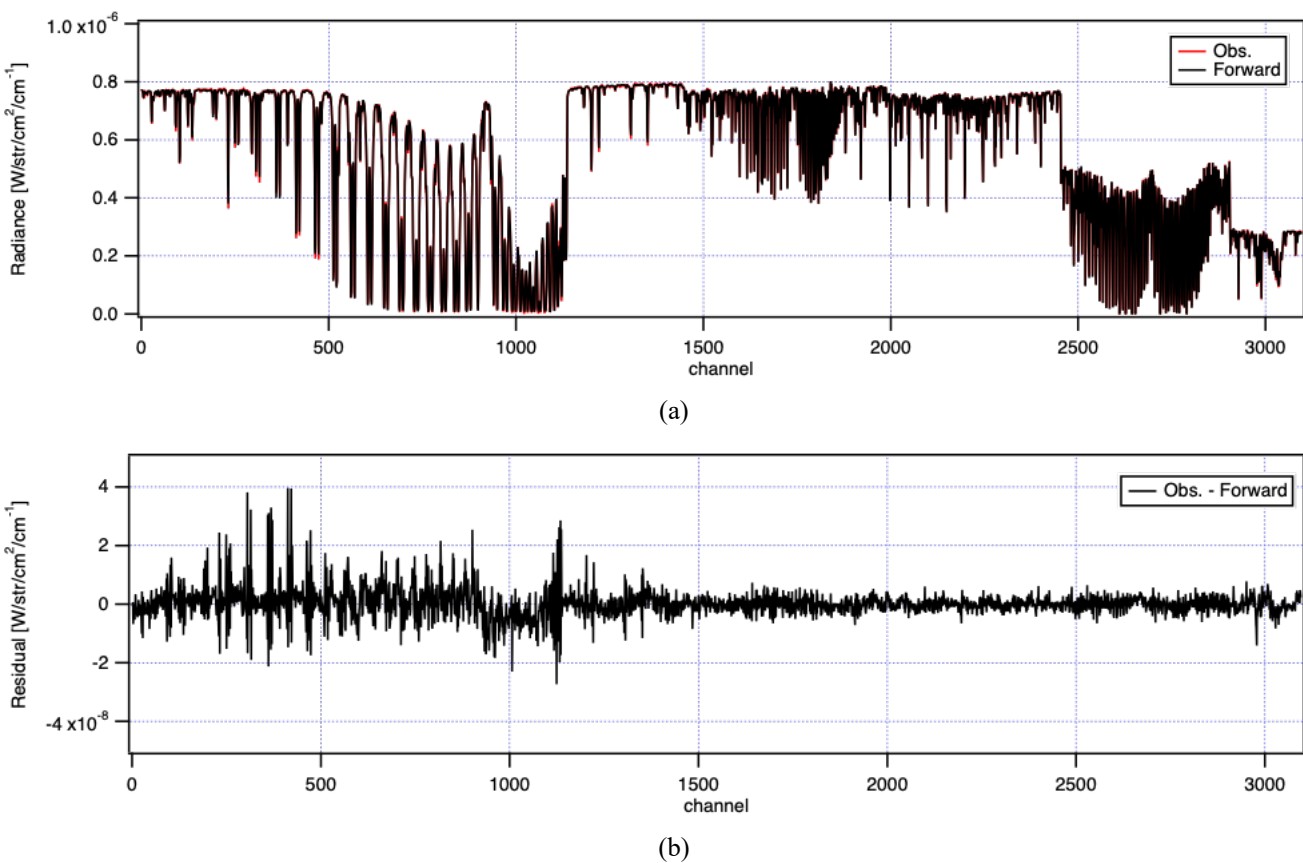

(a)

(b)


**Figure 18.** An example of residual spectra between observed and forward calculated over vicarious calibration site (RRV) on July 1, 2019; (**a**) difference between the observed spectra and the forward calculation; (**b**) the residual between the observed spectra and the forward calculation.



**Figure 19.** A global map for cloud cover index lower than 1 % on September 2019. The top and bottom figures shows before and after intelligent pointing, respectively.
