# Peer review of "Thermal and near-infrared sensor for carbon observation Fouriertransform spectrometer-2 (TANSO-FTS-2) on the Greenhouse Gases Observing Satellite-2 (GOSAT-2) during its first year on orbit"

_Atmospheric Measurement Techniques, 2020_

## Referee Comment (RC1) · Anonymous Referee #1 · 31 Oct 2020

General Comments:

The authors have provided a very thorough description of the mission, instrument, and operations process. Both successes and challenges are presented. For several quantities that are listed, it was not always clear what the target/threshold performance level is, or what the consequence was when it was not achieved.

Specific Comments:

[Figure]

Line 84: why do the spectral ranges of the forward and backward channels of CAI-2 differ? Line 182: were CO2 or CH4 also detectable? Line 283: at what radiance levels does saturation occur? How often is the saturation flag set? Line 546: Instead of "slightly wider", quantify the typical difference in ILS width Table 5: why are separate wavenumbers listed for s & p when they're always the same? Fig 5d: why does Band 5 have a linear relationship between SNR and Radiance while the other bands show a square root dependence? Fig 18b: This would be more informative if the bands were split and the residuals were not in absolute radiance units, but relative to the continuum signal

Technical Corrections: Capitalize "Earth" Use greek mu ($\mu$) instead of "u" for micrometer use "sr" for steradian instead of "str" Line 371: reword "emissivity presents higher" Line 590: "increased by 1.7" -> "increased by a factor of 1.7" Fig 7a: check punctuation in flowchart Fig 9 caption: check spacing between (a) and the panel description

---

## Referee Comment (RC2) · Ray Nassar (Referee) · 29 Dec 2020

General Comments

"Thermal and near-infrared sensor for carbon observations Fourier transform spectrometer-2 (TANSO-FTS-2) on the Greenhouse Gases Observing Satellite-2 (GOSAT-2) during its first year on orbit by Suto et al. describes the on-orbit performance of the TANSO-FTS-2, launched in October 2018. Due to the similarities be-

tween GOSAT and GOSAT-2, much of the paper clarifies the differences in design and performance relative to TANSO-FTS on GOSAT, which is useful for readers interested in calibration issues or simply for informed data users. However, one truly novel aspect of GOSAT-2, which is not described anywhere else in the scientific literature is intelligent pointing. I think it would be valuable to expand this brief section of the manuscript with some more detail about the intelligent pointing approach and implementation as well as the results. With such as large fraction of data lost due to clouds with standard observing (and post-processing cloud filters), the factor of 1.8 improvement in coverage is interesting, yet potential for further improvement can only be assessed if a little more detail were to be provided. Overall, with this issue addressed and the specific points below, this paper should serve as a useful reference for the performance of TANSO-FTS on GOSAT-2.

Specific Comments and Technical Corrections

Line 1-2: Should not have dash on "Fourier-transform" in title.

Line 21-23 and throughout paper: Greek letter mu should be used instead of u.

Line 23: 0.20 cm-1 is the spectral sampling interval, while the spectral resolution of the instrument line shape relates to the observed full width at half maximum and will be at least 1.2 times the spectral sampling interval.

Line 38: "UNFCCC" not "UNFCC"

Line 75: It would be useful if the authors would comment on the mass and/or size of the spacecraft since there is nothing in Figure 1 to allow a reader to gauge the scale, for example, no person in the image.

Line 81 and throughout: Should replace "deg" with ° throughout the paper

Line 93: Please clarify if the 'turnaround time' is for the interferometer scan arm or if it is related to pointing.
Line 120: It would be useful to state either the size of one CMOS camera pixel on the ground or the full field of view size. Furthermore, what wavelength range does the CMOS camera cover?

Line 125: "effective aperture size" is stated but is this somehow different from the true aperture size?

Line 140: Should read "9.6 km diameter"

Line 144: Rather than repeating the spectral regions in the text when they are already in Table 2, the text should just refer to Table 2. It would be useful if Table 2 also listed the GOSAT bands for comparison as was done with orbits.

Line 215: The SNRs between the S and P polarizations are surprisingly different. Is this due to the detectors not being identical or something else?

Line 273: Mentions nadir and many calibration modes. What about glint? Does nadir here actually mean any Earth scene observations?

Line 331: phase-corrected

Line 341: "gauss" should be "Gaussian"

Line 485/504: Temperature difference should not be given in %. It is much better to use absolute units (K).

Line 517: While the small mean offset is encouraging, the authors should comment on the factors contributing to the standard deviation of the offset (0.17 km latitude and 0.18 km longitude) which is not entirely negligible.

Line 529: 0.2 cm-1 is the spectral sampling interval (see comment for line 23)

Line 529-539, Figure 6 and Figure 17 – A band (12950 and 13250 cm-1) is very asymmetric. Is this mainly attributed to optical misalignment and was this present in pre-launch testing or only present on-orbit?

Line 552: Is the same CMOS camera mentioned earlier used for the cloud identification or is this a different camera? This should be clarified in the text. In either case, some details should be provided like spatial resolution and FOV dimensions. How long does it take to process the image onboard in real time?

Line 560-564: While S, M and V are defined by equations 13-15, do they have any descriptive interpretation. What are units or typical range of the raw pixel measures?

Line 635-714: References should be listed alphabetically, but O'Brien et al and Parker et al are not.

Table 5. Table should be simplified since the wavenumber for the p and s polarizations of each band is exactly the same (to the precision given).

Table 7. It should be specified that this is the ratio of radiance for GOSAT-2/GOSAT.

Figure 14 – It would be better to say "AIRS" than "Aqua" since each Aqua instrument would have a different viewing pattern.

Figure 18 – Residuals in oxygen band seem indicative of a poor fit. The authors should comment on this.

---

## Author Comment (AC2) · 29 Jan 2021

**Response to referee comments on manuscript amt-2020-360**

First of all, we would like to thank the referee Dr. Ray Nassar for his constructive comments, which helped to improve the manuscript. We replied all comments and questions as follows. The referee's comments are in blue text.

General Comments

**Referee:**

"Thermal and near-infrared sensor for carbon observations Fourier transform spectrometer-2 (TANSO-FTS-2) on the Greenhouse Gases Observing Satellite-2 (GOSAT-2) during its first year on orbit" by Suto et al. describes the on-orbit performance of the TANSO-FTS-2, launched in October 2018. Due to the similarities between GOSAT and GOSAT-2, much of the paper clarifies the differences in design and performance relative to TANSO-FTS on GOSAT, which is useful for readers interested in calibration issues or simply for informed data users. However, one truly novel aspect of GOSAT-2, which is not described anywhere else in the scientific literature is intelligent pointing. I think it would be valuable to expand this brief section of the manuscript with some more detail about the intelligent pointing approach and implementation as well as the results. With such as large fraction of data lost due to clouds with standard observing (and post-processing cloud filters), the factor of 1.8 improvement in coverage is interesting, yet potential for further improvement can only be assessed if a little more detail were to be provided. Overall, with this issue addressed and the specific points below, this paper should serve as a useful reference for the performance of TANSO-FTS on GOSAT-2.

**Author's reply:**

Thank you very much for reviewing our manuscript. We revised our manuscript with changes tracked.

As you suggested, we added more detail explanation for intelligent pointing specification, functionality and performance in the revised manuscript.

We revised Fig. 19 to present clear difference between before and after intelligent pointing effect in the world view. In the specific region, we added Fig. 20, which support to clear understand the impact of intelligent pointing.

Figure 21 presents the typical images for positive and negative cases of unsuccessful intelligent pointing as well as successful intelligent pointing

Specific Comments and Technical Corrections:

**Referee:**

Line 1-2: Should not have dash on "Fourier-transform" in title.

**Author's reply:**

We removed the dash on "Fourier transform" in title.

**Referee:**

Line 21-23 and throughout paper: Greek letter mu should be used instead of u.

**Author's reply:**

We corrected the Greek letter mu instead of u.

**Referee:**

Line 23: 0.20 cm$^{-1}$ is the spectral sampling interval, while the spectral resolution of the instrument line shape relates to the observed full width at half maximum and will be at least 1.2 times the spectral sampling interval.

**Author's reply:**

We corrected the wording related to the spectral sampling interval in both, the text and Table 2.

**Referee:**

Line 38: "UNFCCC" not "UNFCC"

**Author's reply:**

We corrected the word of "UNFCCC" instead of "UNFCC".

**Referee:**

Line 75: It would be useful if the authors would comment on the mass and/or size of the spacecraft since there is nothing in Figure 1 to allow a reader to gauge the

scale, for example, no person in the image.

**Author's reply:**

We added the scale information on Fig. 1. In addition, the size and mass of the spacecraft are added in Table 1. We also added the size and mass for GOSAT, to allow for comparison with GOSAT-2.

[Figure]

**Figure 1.** Photograph of GOSAT-2 before launch.

**Table 1.** Satellite and orbit parameters of GOSAT-2 and GOSAT.

| Specification Items | | GOSAT-2 Specifications | GOSAT-2 Remarks | GOSAT Specifications | GOSAT Remarks |
|---|---|---|---|---|---|
| SAT | Size (H x W x D) | 5.8m x 2.0m x 2.1m | | 3.7m x 1.8m x 2.0m | |
| | Paddle span | 16.5m | | 13.7m | |
| | Weight | < 1800kg | | < 1750kg | |
| | Power generation | 5kW | | 3.8kW | |
| O R B I T | Type | Sun synchronous, quasi-recurrent | | Sun synchronous, quasi-recurrent | |
| | Local overpass time | 13hours±15minutes | Descending node | 13hours±15minutes | Descending node |
| | Altitude | 612.98km | Not including altitude variations in orbit | 666 ± 0.6km | |
| | Inclination angle | 97.84° | | 98.0° ± 0.1° | |
| | Eccentricity | 0.00106 | Frozen orbit | | Frozen orbit |
| | Period | Approximately 98.1minutes | | Approximately 98.1minutes | |
| | Repeat cycle | 6days (89paths) | | 3days (44paths) | |
| | Origin point | An orbit exactly over Lamont, OK (Latitude 36.6North, Longitude 97.5 West) | | An orbit exactly over Tsukuba, Ibaraki (Latitude 36.1North, Longitude 140.1 East) | |
| | Descending node accuracy | ±2.5km | Depending upon the frequency of orbit control manuevers | ±2.5km | Depending upon the frequency of orbit control manuevers |

]

**Referee:**

Line 81 and throughout: Should replace "deg" with ° throughout the paper

**Author's reply:**

We changed "deg" to "°" in the revised manuscript.

**Referee:**

Line 93: Please clarify if the 'turnaround time' is for the interferometer scan arm or if it is related to pointing.

**Author's reply:**

The turnaround time includes both, the interferometer scan arm turnaround motion time and the time required for pointing. There is also allowance for accurate measurement

timing. In other words, it is the time between one observation and the next observation. When intelligent pointing operations are requested, changing the pointing location, taking image, identifying cloud location and repointing are performed during the turnaround duration.

We added the following sentence:

"which includes changing the pointing location, taking image, identifying cloud location in the image and repointing to cloud free location as well as time to insure precise measurement timing."

**Referee:**

Line 120: It would be useful to state either the size of one CMOS camera pixel on the ground or the full field of view size. Furthermore, what wavelength range does the CMOS camera cover?

**Author's reply:**

The full field of view size is 30 km by 50 km for the along-track by cross-track direction. We modified the sentence as follows:

" One beam is directed to the CMOS video camera (608 x 1024 pixels, which corresponds to ~0.1 km spatial resolution with 30 km in along-track and 50 km in cross-track coverage) for identifying the scene image and the second beam is introduced to the interferometer (FTS). The camera image is also used to identify cloud positions in the field of view. The camera has a red, green, blue detection capability with 8-bit digitalization range where red corresponds to the band 575-750 nm, green to the band 500-575 nm, and blue to the band 400-500 nm. "

**Referee:**

Line 125: "effective aperture size" is stated but is this somehow different from the true aperture size?

**Author's reply:**

The diameter of a cube corner is 77 mm which forms the geometric aperture. However, due to refraction effects by the beam splitters, the combined beam size of interferometry should be narrower than the actual cube corner size. Considering this effect, the effective aperture size of the FTS-2 becomes 73 mm. To avoid confusion, we modified the sentence as follows:

" The aperture diameter is 73 mm and is defined at the cube corner mirrors of FTS-2, "

**Referee:**

Line 140: Should read "9.6 km diameter"

**Author's reply:**

We removed the word of "in" before diameter.

**Referee:**

Line 144: Rather than repeating the spectral regions in the text when they are already in Table 2, the text should just refer to Table 2. It would be useful if Table 2 also listed the GOSAT bands for comparison as was done with orbits.

**Author's reply:**

We modified the text and referred to Table 2 for the spectral regions. Also, we modified Table 2, and added the GOSAT specifications to support a better understanding of the difference between GOSAT and GOSAT-2.

**Table 2.** Spectroscopic specifications of the TANSO-FTS-2.

| | Band 1 | | Band 2 | | Band 3 | | Band 4 | | Band 5 | |
|---|---|---|---|---|---|---|---|---|---|---|
| | G2* | G** | G2 | G | G2 | G | G2 | G | G2 | G |
| Spectral Coverage (cm$^{-1}$) | 12950-13250 | 12900-13200 | 5900-6400 | 5800-6400 | 4200-5200 | 4800-5200 | 1188-1800 | 700-1800 | 700-1188 | N/A |
| Polarization Obs. | 2 | | 2 | | 2 | | N/A | | N/A | |
| Spectral Sampling Interval (cm$^{-1}$) | 0.2 cm$^{-1}$(Both sides scan) (MOPD +/-2.5 cm) | | | | | | | | | |
| Sampling Number | 153090 | 65536 | 76545 | 65536 | 76545 | 65536 | 38250 | 38186 | 38250 | N/A |
| Full Width Half Maximum (cm$^{-1}$) | < 0.4 | | < 0.27 | | < 0.27 | | < 0.27 | | < 0.27 | |
| Detector | Si | Si | PV-MCT | InGaAs | PV-MCT | InGaAs | PV-MCT | PC-MCT | PC-MCT | N/A |

*G2: GOSAT-2

**G: GOSAT

**Referee:**

Line 215: The SNRs between the S and P polarizations are surprisingly different. Is this due to the detectors not being identical or something else?

**Author's reply:**

Detectors for P and S polarization bands are the same. However, the polarization sensitivities of the beam splitters, made of ZnSe, are largely different between P and S which causes the significant difference of transmittance. We added the following explanation for the difference of P and S polarization sensitivity.

"In the case of FTS-2, the SNR for s-polarization is higher than that of p-polarization and is related to the total polarization sensitivity in ZnSe beam splitters."

**Referee:**

Line 273: Mentions nadir and many calibration modes. What about glint? Does nadir here actually mean any Earth scene observations?

**Author's reply:**

The glint observations are categorized as nadir observations. For GOSAT, the nadir observations are categorized in two separate modes: one is grid mode, the other is target mode including glint observations. For GOSAT-2, all of observations are freely programmable. So, we do not apply the separate names for Earth scene observations. However, we still report a glint flag in the Level 1 files to identify glint observations.

As for the calibration mode, we create a separate L1 files for black body, deep space, solar, lunar, ILS, and Dark calibrations. The details of the file format are described in the GOSAT-2 Level 1 Data Description Document for TANSO-FTS-2.

To clarify the difference of the L1 file management between GOSAT and GOSAT-2, we added the following sentence:

"In contrast to GOSAT, any Earth scene observations, be it grid observations, target observations or glint observation are included in the same L1 file with land/glint flags (GOSAT-2 Level-1 Data Description Document for TANSO-FTS-2, 2020)."

**Referee:**

Line 331: phase-corrected

**Author's reply:**

We corrected the word.

**Author's reply:**

We corrected the word.

Referee:

Line 485/504: Temperature difference should not be given in %. It is much better to use absolute units (K).

**Author's reply:**

We modified the temperature difference unit in (K).

Referee:

Line 517: While the small mean offset is encouraging, the authors should comment on the factors contributing to the standard deviation of the offset (0.17 km latitude and 0.18 km longitude) which is not entirely negligible.

**Author's reply:**

As we described in the text, the geometric characterization is performed with reference to ground control points. This technique is the same as for GOSAT geometric characterization. During the image motion compensation operations by the pointing mirror, we simultaneously acquire the set-point pointing angles for along-track and cross- track as well as the corresponding measured angles with a sampling of 100Hz. By comparing the set-point and measured angles, we can calculate the pointing fluctuation which we found negligible. Thus, we conclude that the standard deviation of the offset is caused by the uncertainty of the geometric characterization method. When we applied the same method to GOSAT, we find the same standard deviation of 0.2 km in both latitude and longitude. Overall, this deviation corresponds to less than 2 % of the FOV size.

Referee:

Line 529: 0.2 cm-1 is the spectral sampling interval (see comment for line 23)

**Author's reply:**

We modified the wording as you suggested including Table 2.

**Referee:**

Line 529-539, Figure 6 and Figure 17 – A band (12950 and 13250 cm-1) is very asymmetric. Is this mainly attributed to optical misalignment and was this present in pre- launch testing or only present on-orbit?

**Author's reply:**

During the pre-launch test phase, we observed the asymmetric ILS. In addition, after the launch, the FWHM was found wider by +0.03cm$^{-1}$. We changed the sentence as follows;

"The best-estimated ILS function for band 1 is slightly wider than that of the prelaunch test. The difference of the FWHM between the prelaunch test and the orbit best-estimated one is found +0.03cm$^{-1}$.  However, a time-dependent term is not implemented in the current best-estimated ILS function."

**Referee:**

Line 552: Is the same CMOS camera mentioned earlier used for the cloud identification or is this a different camera? This should be clarified in the text. In either case, some details should be provided like spatial resolution and FOV dimensions. How long does it take to process the image onboard in real time?

**Author's reply:**

It is the same camera as mentioned in section 3.1. The pointing relocation including the cloud identification process is performed within a total of 0.65 sec. For real-time onboard processing, image detection and cloud identification is processed within 0.2 sec as designed. We added the following sentence:

"Both image detection and cloud identification are processed within 0.2 sec. "
  We added more detail for this camera in section 3.1

"One beam is directed to the CMOS video camera (608 x 1024 pixels, which corresponds to ~0.1 km spatial resolution with 30 km in along-track and 50 km in cross-track coverage) for identifying the scene image and the second beam is introduced to the interferometer (FTS). The camera image is also used to identify cloud positions in the field of view. The camera has a red, green, blue detection capability with 8-bit digitalization range where red corresponds to the band 575-750 nm, green to the band 500-575 nm, and blue to the band 400-500 nm. "

**Referee:**

Line 560-564: While S, M and V are defined by equations 13-15, do they have any descriptive interpretation. What are units or typical range of the raw pixel

measures?

**Author's reply:**

Due to the limitation of on-board processing resources, our cloud determination approach is based on based on simple brightness and chroma thresholds.

To keep on-board processing time, the acquired images are directly used in this calculations. In other words, the digital numbers (DN), which are corresponding read, green, and blue bands, are directory used in the processing. For the typical cloud free observations over city, S, M, and V are corresponded 10-40 DN, 50-70 DN, and 10-40 DN, respectively.

We added more detail explanation as follows;

"Each of the color composites has 8-bit digitalization. Thus, $S$, $M$, and $V$ vary across the 0-255 range. For typical cloud free observations over city areas, the composite values $S$, $M$, and $V$ are roughly in the range 10-40 DN, 50-70 DN, and 10-40 DN, respectively. "

**Referee:**

Line 635-714: References should be listed alphabetically, but O'Brien et al and Parker et al are not.

**Author's reply:**

We corrected the references.

**Referee:**

Table 5. Table should be simplified since the wavenumber for the p and s polarizations of each band is exactly the same (to the precision given).

**Author's reply:**

We modified  Table 5.

**Table 5.** Wavenumbers considered for the radiometric performance analyses.

| Color | Band 1p & 1s | Band 2p & 2s | Band 3p & 3s |
|---|---|---|---|
| red | 12975.2 cm$^{-1}$ | 6174.7 cm$^{-1}$ | 4808.8 cm$^{-1}$ |
| orange | 12993.8 cm$^{-1}$ | 6186.8 cm$^{-1}$ | 4822.9 cm$^{-1}$ |
| green | 13027.6 cm$^{-1}$ | 6229.6 cm$^{-1}$ | 4849.9 cm$^{-1}$ |
| blue | 13122.1 cm$^{-1}$ | 6257.8 cm$^{-1}$ | 4871.4 cm$^{-1}$ |
| purple | 13171.8 cm$^{-1}$ | 6277.9 cm$^{-1}$ | 4880.4 cm$^{-1}$ |

**Referee:**

Table 7. It should be specified that this is the ratio of radiance for GOSAT-2/GOSAT.

**Author's reply:**

We modified the text and Table 7 title as "Ratio of radiances between TANSO-FTS and TANSO-FTS-2"

**Table 7.** Ratio of radiances between TANSO-FTS and TANSO-FTS-2 in SWIR region

| Band 1p | | Band 1s | | Band 2p | | Band 2s | | Band 3p | | Band 3s | |
|---|---|---|---|---|---|---|---|---|---|---|---|
| Ave. | Std. | Ave. | Std. | Ave. | Std. | Ave. | Std. | Ave. | Std. | Ave. | Std. |
| 12975.2 cm$^{-1}$ | | 12975.2 cm$^{-1}$ | | 6174.7 cm$^{-1}$ | | 6174.7 cm$^{-1}$ | | 4808.8 cm$^{-1}$ | | 4808.8 cm$^{-1}$ | |
| 1.011 | 0.314 | 0.987 | 0.325 | 1.000 | 0.376 | 1.01 | 0.332 | 1.021 | 0.479 | 1.022 | 0.48 |
| 12993.8 cm$^{-1}$ | | 12993.8 cm$^{-1}$ | | 6186.8 cm$^{-1}$ | | 6186.8 cm$^{-1}$ | | 4822.9 cm$^{-1}$ | | 4822.9 cm$^{-1}$ | |
| 1.013 | 0.315 | 0.995 | 0.328 | 1.003 | 0.377 | 1.01 | 0.333 | 1.03 | 0.484 | 1.033 | 0.485 |
| 13027.6 cm$^{-1}$ | | 13027.6 cm$^{-1}$ | | 6229.6 cm$^{-1}$ | | 6229.6 cm$^{-1}$ | | 4849.9 cm$^{-1}$ | | 4849.9 cm$^{-1}$ | |
| 1.001 | 0.311 | 0.994 | 0.327 | 1.000 | 0.376 | 1.003 | 0.33 | 1.015 | 0.477 | 1.021 | 0.479 |
| 13122.1 cm$^{-1}$ | | 13122.1 cm$^{-1}$ | | 6257.8 cm$^{-1}$ | | 6257.8 cm$^{-1}$ | | 4871.4 cm$^{-1}$ | | 4871.4 cm$^{-1}$ | |
| 0.984 | 0.306 | 0.982 | 0.323 | 0.997 | 0.375 | 0.995 | 0.327 | 0.987 | 0.464 | 0.995 | 0.467 |
| 13171.8 cm$^{-1}$ | | 13171.8 cm$^{-1}$ | | 6277.9 cm$^{-1}$ | | 6277.9 cm$^{-1}$ | | 4880.4 cm$^{-1}$ | | 4880.4 cm$^{-1}$ | |
| 0.985 | 0.306 | 0.976 | 0.322 | 1.001 | 0.376 | 0.997 | 0.328 | 1.016 | 0.477 | 1.021 | 0.479 |
| Total | | | | | | | | | | | |
| 0.999 | 0.31 | 0.987 | 0.325 | 1.000 | 0.376 | 1.003 | 0.33 | 1.014 | 0.476 | 1.018 | 0.478 |

**Referee:**

Figure 14 – It would be better to say "AIRS" than "Aqua" since each Aqua instrument would have a different viewing pattern.

**Author's reply:**

We corrected the caption.

[Figure]

**Figure 14.** Inter comparison between TANSO-FTS-2 and others; coincident latitude and longitude map

between TANSO-FTS-2, AIRS (o) and IASI (x).

**Referee:**

Figure 18 – Residuals in oxygen band seem indicative of a poor fit. The authors should comment on this.

**Author's reply:**

We added the following sentences;

"As figure 18 suggests, substantial spectral residuals remain in the oxygen band. This indicates that we require improved knowledge on the ILS function, especially for band 1. We anticipate that a time-dependent term for the ILS function will be a key step forward to improve the fitting. In a future calibration update, the time-dependency will be implemented."

End of document

---

## Author Response (AR1)

**Authors response to referee comments on revised version of manuscript**

"Thermal and near-infrared sensor for carbon observation Fourier transform spectrometer-2 (TANSO-FTS-2) on the Greenhouse Gases Observing Satellite-2 (GOSAT-2) during its first year on orbit " on Hiroshi Suto et al., amt-2020-360

**Dear Editor,**

We thank the editor and referees for their careful review of our manuscript and thoughtful comments. We modified the manuscript as the referee suggested. We have uploaded (1) author's response, (2) two PDF files with and without changes tracked, and manuscript in MS-word with changes tacked.

**Response to referee comments on manuscript amt-2020-360**

First of all, we would like to thank referee #1 for his/her constructive comments, which helped us to improve the manuscript. We replied all comments and questions as follows. The referee's comments are in blue text.

**Anonymous Referee #1**

General Comments:

**Referee:**

The authors have provided a very thorough description of the mission, instrument, and operations process. Both successes and challenges are presented. For several quantities that are listed, it was not always clear what the target/threshold performance level is, or what the consequence was when it was not achieved.

**Author's reply:**

Thank you very much for reviewing our manuscript. We revised our manuscript with changes tracked.

Specific Comments:

**Referee:**

Line 84: why do the spectral ranges of the forward and backward channels of CAI-2 differ?

**Author's reply:**

The cloud detection is performed in the 674, 869 and 1630 nm bands, which are common to the forward and backward direction. The cloud detection algorithm is described in Ishida et al (2009) and Oishi et al (2018). In the operational XCO2 and XCH4 retrieval algorithm, developed by NIES, the cloud location information from CAI-2 is used to pre-screen cloudy observation scenes. The three spectral bands are mounted on both, the forward and backward looking directions to avoid missing any spatial cloud information.

The four UV and VIS bands are not used in GOSAT-2 operational cloud detection processing. Their wavelengths have been chosen after consultation of the Japanese science community and in the view of the available GOSAT-2 system resources. We added the additional two references for CAI-2 in the manuscript.

Ishida, H., and Nakajima, T.Y.: Development of an unbiased cloud detection algorithm for a spaceborne multispectral imager, J. Geophys. Res., 114, D07206, doi:10.1029/2008JD010710, 2008.

Oishi, Y., Ishida, H., Nakajima, T. Y., Nakamura, R., Matsunaga, T.: The impact of

different support vectors on GOSAT-2 CAI-2 L2 Cloud disclination., Remote Sens. 2017, 9, 1236; doi:10.3390/rs9121236, 2017.

**Referee:**

Line 182: were CO2 or CH4 also detectable?

**Author's reply:**

 $CO_2$  and  $CH_4$  in the 1.6 µm region are not detectable in our pre-launch test configuration. In contrast,  $CO_2$  in the region from 4900 to 5000 cm-1 and water vapor are detectable and cause some interferences when characterizing the signal-to-radiance conversion coefficients. This is a reason why we combined atmospheric tests and thermal vacuum tests.

We corrected the manuscripts as follows;

"Due to the interference of oxygen lines in band 1, water vapor and  $CO_2$  lines in band 3,"

**Referee:**

Line 283: at what radiance levels does saturation occur? How often is the saturation flag set?

**Author's reply:**

Saturation is diagnosed in the interferogram domain. Due to the 14-bit resolution (+/-8191) of the ADCs, the saturation criterion is set as full bit range (+8191) in digital number units. The saturation rate, which is defined as the ratio between the total observation number and the number of saturated observations, during February 2019 to March 2020 is 8.9%, 13.0%, 6.3%, 6.6%, 2.9%, and 3.7% for the b1p, b1s, b2p, b2s, b3p, and b3s detectors, respectively.

We added the following two sentences are added in section 4.1 and 5.1;

In section4.1:

In this case, the saturation criterion is set to +8191 digital number units.

In section5.1

In addition, to minimize the acquisition of useless data, the gain settings for each of the bands are examined during the first year of operation. As a result, the saturation rate, which is defined as the ratio between the total observation number and the number of saturated observations, during February 2019 to March 2020 is 8.9%, 13.0%, 6.3%, 6.6%, 2.9%, and 3.7% for bands 1p, 1s, 2p, 2s, 3p, and 3s detectors, respectively. For bands 4 and 5, there was no saturated data during the considered period. The main reason for saturation is cloudy scene observation, especially in band 1.

**Referee:**

Line 546: Instead of "slightly wider", quantify the typical difference in ILS width

**Author's reply:**

We added the following sentence is added in the manuscript.

"The best-estimated ILS function for band 1 is slightly wider than that of the prelaunch test. The difference of the FWHM between the prelaunch test and the orbit best-estimated one is found +0.03 cm-1. However, a time-dependent term is not implemented in the current best-estimated ILS function."

**Referee:**

Table 5: why are separate wavenumbers listed for s & p when they're always the same?

**Author's reply:**

As you suggested, wavenumbers for analyses are always the same for both, p and s bands. We modified the Table 5 as follows;

| Color  | Band 1p & 1s             | Band 2p & 2s            | Band 3p & 3s            |
|--------|--------------------------|-------------------------|-------------------------|
| red    | 12975.2 cm -1 | 6174.7 cm -1 | 4808.8 cm -1 |
| orange | 12993.8 cm -1 | 6186.8 cm -1 | 4822.9 cm -1 |
| green  | 13027.6 cm -1 | 6229.6 cm -1 | 4849.9 cm -1 |
| blue   | 13122.1 cm -1 | 6257.8 cm -1 | 4871.4 cm -1 |
| purple | 13171.8 cm -1 | 6277.9 cm -1 | 4880.4 cm -1 |

Table 5. Wavenumbers considered for the radiometric performance analyses.

**Referee:**

Fig 5d: why does Band 5 have a linear relationship between SNR and Radiance while the other bands show a square root dependence?

**Author's reply:**

In the case of photon shot noise dominating, the relationship between SNR and radiance shows a square root dependence. The band 5 has electronic noise that exceeds the photon-dependent noise. As described in the manuscript, the non-linearity correction for band 5 requires improvements. If the non-linearity correction is not perfect, it also contributes to the distortion of the relationship between SNR and radiance. We believe that this finding indicates that calibration of band 5 needs to be revisited.

**Referee:**

Fig 18b: This would be more informative if the bands were split and the residuals were not in absolute radiance units, but relative to the continuum signal.

**Author's reply:**

We split the Figure 18 in the individual bands, but we kept the display of residuals in absolute radiance units. Since our spectra cover optically thick absorption lines and since FTS-2 has high spectral resolution, radiances become very small for some spectral samples. Relative residuals would be dominated by these samples (division by small number) and would mask the rest of the residuals.